# Role of the Ca^2+^-ATPase Pump (SERCA) in Capacitation and the Acrosome Reaction of Cryopreserved Bull Spermatozoa

**DOI:** 10.3390/cells14231892

**Published:** 2025-11-28

**Authors:** Maia A. Rodríguez, Alejandro Orlowski, Enrique L. Portiansky, Paola Ferrero

**Affiliations:** 1Reprosemyx S.A., Calle 4 # 693, La Plata 1900, Argentina; 2Laboratorio de Análisis de Imágenes, Universidad Nacional de La Plata, La Plata 1900, Argentina; 3Centro de Investigaciones Cardiovasculares “Dr. Horacio E. Cingolani”, Facultad de Ciencias Médicas, Universidad Nacional de La Plata—Consejo Nacional de Investigaciones Científicas y Técnicas, Av 60 & 120 2do piso, La Plata 1900, Argentina; 4Departamento de Ciencias Básicas y Experimentales, Escuela de Ciencias Agrarias, Naturales y Ambientales, Universidad Nacional del Noroeste de la Provincia de Buenos Aires (UNNOBA), Pergamino 2772, Argentina

**Keywords:** calcium, SERCA, cryopreserved sperm, bull, actin, acrosome, capacitation

## Abstract

Cryopreservation of sperm is a resource used for artificial insemination. In the case of bovines, it ensures the reproduction of animals through cells from males with a highlighted genetic background. The fertilisation capacity of sperm cells is achieved after capacitation, a process that includes subcellular modifications that could be altered by cryopreservation. Intracellular calcium handling plays a crucial role in the development of capacitation, culminating in the acrosomal reaction. SERCA protein is responsible for calcium reuptake into the acrosome, which is one of the main calcium reservoirs of sperm cells. In this work, we studied the relationships between SERCA activity and sperm motility, capacitation progression, actin polymerisation, and intracellular calcium handling in cryopreserved bovine spermatozoa. Inhibition of SERCA activity reduced sperm motility and induced hyperactivation patterns. It also increased the proportion of cells with acrosomal reaction and earlier actin depolymerisation, an event necessary to induce the acrosomal reaction. All changes occurred in concordance with a significant increase in intracellular calcium concentration (Ca^2+^). Our findings suggest that a thapsigargin-sensitive Ca^2+^ pump consistent with SERCA activity remains responsive in cryopreserved spermatozoa from the bull studied, under specific cryopreservation and incubation conditions tested, and may contribute to Ca^2+^ handling, motility changes, and premature acrosomal exocytosis.

## 1. Introduction

Sperm capacitation is a crucial process required for the acquisition of fertilising capacity in spermatozoa, enabling them to interact appropriately with the oocyte. It involves structural and functional modifications that occur as spermatozoa transit through the female reproductive tract or, in vitro, during incubation in capacitation media [1]. The changes associated with capacitation include alterations in the plasma membrane and cell surface properties resulting from protein desorption and redistribution, modifications in phospholipid organisation, cholesterol depletion, protein phosphorylation, alterations in membrane potential and intracellular pH, and an increase in intracellular calcium concentration [2]. These early events, as part of the capacitation process, ultimately lead to the acrosome reaction [3]. Capacitation has been extensively investigated in numerous mammalian species [2,4,5,6] and is of relevance for understanding fertilisation mechanisms and optimising in vitro fertilisation protocols.

The regulation of intracellular calcium (Ca^2+^_i_) during capacitation by several channels and pumps, such as CatSper, SPCA, and PMCA, influences both flagellar motility and the acrosomal reaction [7,8,9]. In addition to the midpiece, plasma membrane proteins and those located in the acrosome help regulate the concentration of this ion within different cellular compartments. A key component in cytosolic Ca^2+^ regulation is SERCA, the principal protein responsible for Ca^2+^ reuptake into the acrosome [10,11,12]. In humans and pigs, SERCA2 is predominantly localised in the acrosome and midpiece [10,12,13]. Although Ca^2+^ regulation occurs in sperm cells across various species, comparatively little information is available regarding this process in cryopreserved bovine spermatozoa.

In addition to modulating motility and the acrosomal reaction, the increase in intracellular calcium (Ca^2+^_i_) promotes actin polymerisation. Actin is a structural protein that forms part of the cytoskeleton and plays a crucial role in cellular reorganisation. During its remodelling, globular actin (G-actin) is converted into filamentous actin (F-actin). In the sperm tail, this transformation contributes to the regulation of motility by stabilising the flagellum and enabling hyperactivation, thereby enhancing the sperm’s ability to move vigorously towards the oocyte [4]. In the sperm head, polymerisation followed by depolymerisation is thought to facilitate the acrosomal reaction [5,6,7,8,9,10,11,12,13,14,15,16,17]. Actin polymerisation and depolymerisation are conserved processes observed across several species, including bovine, murine, and human spermatozoa [15,18].

All these tightly regulated events that occur under physiological conditions may be affected by in vitro manipulation. For instance, they may differ between freshly ejaculated and cryopreserved spermatozoa. Cryopreservation compromises membrane integrity, induces nuclear fragmentation, and increases the production of reactive oxygen species (ROS) and cellular necrosis [19]. Moreover, cytoplasmic Ca^2+^ levels are elevated in cryopreserved sperm, promoting capacitation (cryo-capacitation) and triggering premature acrosomal reactions [20]. Consequently, cryopreservation may alter subcellular processes. We hypothesised that inhibition of SERCA would disrupt Ca^2+^ regulation and modify capacitation dynamics in cryopreserved bovine spermatozoa. Therefore, in the present study, we investigated the role of SERCA in cryopreserved bull sperm by analysing motility, the acrosomal reaction, Ca^2+^ regulation, and actin polymerisation.

## 2. Materials and Methods

### 2.1. Thawing of Cryopreserved Bovine Spermatozoa

Cryopreserved bovine spermatozoa from Angus bulls were obtained from C.I.A.V.T. Sperm Production Limited Cooperative (Venado Tuerto, Argentina). The purchased straws originated from a single ejaculate of one bull and were aliquoted into 250 µL samples corresponding to the same production batch. On the day of the experiment, the samples were thawed in a water bath at 38 °C and resuspended in 8 mL of non-capacitating TALP solution containing 100 mM NaCl, 3.1 mM KCl, 0.26 mM NaH_2_PO_4_·H_2_O, 0.4 mM MgCl_2_·6H_2_O, 2.7 mM CaCl_2_, 25 mM NaHCO_3_, 43 mM lactate, 1 mM pyruvate (all reagents from Biopack Inc., Ciudad Autónoma de Buenos Aires, Argentina), and 10 mM HEPES (Roth Inc., Karlsruhe, Germany), ref. [21] in order to remove the freezing medium. Although this formulation contains bicarbonate, which may initiate some early capacitation-related events, it does not support the complete cascade leading to capacitation or the acrosome reaction within the short incubation period used here and is therefore considered functionally non-capacitating in this experimental context. The samples were then centrifuged at 300× *g* for 5 min, and the resulting pellet was resuspended in 1 mL of capacitating TALP medium (100 mM NaCl, 3.1 mM KCl, 0.26 mM NaH_2_PO_4_·H_2_O, 0.4 mM MgCl_2_·6H_2_O, 2.7 mM CaCl_2_, 25 mM NaHCO_3_, 43 mM lactate, 1 mM pyruvate, 10 mM HEPES, and 0.3% BSA). Spermatozoa were maintained at 38 °C until use.

### 2.2. Immunohistochemistry for the Detection of SERCA in Sperm Cells

Thawed spermatozoa were fixed in 4% (*w*/*v*) paraformaldehyde (PFA) for 7 min, permeabilised with 0.2% Triton X-100 for 10 min at −20 °C, and washed three times with phosphate-buffered saline (PBS; 137 mM NaCl, 2.7 mM KCl, 4.3 mM Na_2_HPO_4_, and 1.47 mM KH_2_PO_4_; pH 7.4). A drop of the resuspended cells was placed on a microscope slide, air-dried at room temperature, and rinsed with PBS. The slide was then incubated with a mouse monoclonal primary anti-SERCA antibody (Santa Cruz Inc. (Dallas, TX, USA), F-1: sc-376235; 1:500) in a blocking solution (1% BSA in PBS) for 12 h at 4 °C. The specificity of the anti-SERCA antibody was assessed by bioinformatic and experimental validation. Alignment of the immunogen sequence (residues 2–29) showed 100% identity between the human (UniProt accession P16615; ATP2A2_HUMAN) and bovine (UniProt accession A0AAA9T933; ATP2A2_BOVIN) orthologues, supporting expected cross-reactivity.

After extensive washing with PBS, the cells were incubated with a Cy3-labelled anti-mouse secondary antibody for 1 h at 37 °C. The slides were then washed with PBS, mounted with a drop of glycerol, and covered with a glass coverslip. Fluorescence images were acquired using an LSM 800 Carl Zeiss microscope (Carl Zeiss, Munich, Germany) coupled to an Axio Observer Z1 camera (excitation: 548 nm; emission: 566 nm). Image analysis was performed using ImageJ software 1.54r to quantify the fluorescence intensity in the spermatozoa.

### 2.3. Processing of Samples for Motility Assays, CTC, FIT-PSA, and Phalloidin Staining

Straws (one per experiment) were thawed on the day of the experiment, as previously described, and maintained in an incubator at 38 °C. Aliquots of 100 µL were collected at 0, 10, 20, and 30 min. Following the removal of the sample corresponding to time 0, caffeine (W222402, Sigma Inc., St. Louis, MO, USA) (final concentration: 1 mg/mL) was added to the capacitating TALP medium, (0.3% BSA, 100 mM NaCl, 3.1 mM KCl, 0.26 mM NaH_2_PO_4_·H_2_O, 0.4 mM MgCl_2_·6H_2_O, 2.7 mM CaCl_2_, 25 mM NaHCO_3_, 43 mM lactate, 1 mM pyruvate, and 10 mM HEPES), to induce capacitation [22]. Fifteen minutes later, 4 µM progesterone (cat. 8783, Sigma Inc.) was added to promote the acrosomal reaction.

A fraction of the sample (50 µL) was used as a control, while another fraction (50 µL) was treated with thapsigargin (Tg, T9033, Sigma Inc.; final concentration: 10 µM) for 2 min at each analysed time point. The dose was selected based on previous reports [21,23,24]. Thapsigargin is a SERCA inhibitor that prevents SERCA-mediated transport of cytoplasmic Ca^2+^ into intracellular reservoirs. In spermatozoa, the resulting depletion of Ca^2+^ in these compartments triggers the influx of extracellular Ca^2+^ into the cytoplasm via store-operated channels. Consequently, cytoplasmic Ca^2+^ levels can be elevated through pharmacological intervention [21]. A schematic timeline of the experimental design is shown in Appendix A.

### 2.4. Analysis of Motility

Control and treated sperm samples (10 µL) were collected at each time point, placed on pre-warmed glass slides, and covered with a coverslip. The slides were maintained at 38 °C on a thermostated microscope stage and observed under transmitted light at 100× magnification using an LSM 800 Carl Zeiss microscope coupled to an Axio Observer Z1 camera. Videos were acquired with ZEN Blue software 3.10 for 3–4 s at a frame rate of 63 frames per second. For each time point (0, 10, 20, and 30 min), 4–5 recordings were analysed per experiment. Motility parameters were quantified using the OpenCASA plugin for ImageJ [25].

We determined the percentage of motile and progressively motile spermatozoa, as well as several motility parameters calculated using OpenCASA 2.0: straight-line velocity (VSL), curvilinear velocity (VCL), average path velocity (VAP), linearity (LIN), wobble (WOB), straightness (STR), head lateral displacement amplitude (mean ALH), maximum ALH, beat-cross frequency (BCF), dance (DNC), mean angular displacement (MAD), and fractal dimension (FD).

### 2.5. Analysis of Capacitation by Chlortetracycline (CTC) Staining

Sperm samples were treated with CTC (cat. 17776, Sigma Inc.) as previously described [26]. Briefly, 5 µL of sperm suspension was mixed with 5 µL of 500 µM CTC solution (750 µM chlortetracycline, cat. 4881, Sigma Inc.; 5 mM cysteine, 130 mM NaCl, 20 mM Tris; all reagents from Biopack Inc.) on a slide for 10 s. The samples were then fixed with 12.5% (*v*/*v*) glutaraldehyde (Sigma Inc.) in 1 M Tris. The mixture was placed on a glass slide, covered with a coverslip, and observed using a Carl Zeiss LSM 800 microscope (excitation: 400–440 nm; emission: 470 nm). The number of spermatozoa displaying different staining patterns—non-capacitated, capacitated, or acrosome-reacted—was quantified from the images.

### 2.6. Evaluation of Acrosomal Reaction

Fluorescein-labelled Pisum sativum agglutinin (FITC-PSA; cat. 21761045-1, GlycoMatrix™, Dublin, OH, USA), which binds to glycoproteins in the acrosomal matrix, was used to assess the acrosomal reaction in spermatozoa. Briefly, 5 µL of sperm suspension was spread onto a slide and allowed to air dry. The spermatozoa were then fixed with 4% (*w*/*v*) PFA for 7 min, incubated with 50 µg/mL FITC-PSA for 1 h, and protected from light. Samples were mounted in glycerol and observed using a Carl Zeiss LSM 800 microscope (excitation: 491 nm; emission: 516 nm). The number of spermatozoa displaying different staining patterns—intact acrosome or acrosome-reacted—was quantified from the images.

### 2.7. Assessment of Actin Polymerisation

To assess actin polymerisation, Alexa Fluor™ 594-phalloidin (cat. A12381, Thermo Fisher (Waltham, MA, USA); excitation: 565 nm, emission: 600 nm), which specifically binds to polymerised actin [27], was used. Briefly, 5 µL of the sperm sample was spread onto a glass slide and air-dried. The samples were fixed with 4% PFA for 7 min and washed three times with PBS for 5 min each. The slides were then immersed in 100% ethanol at −20 °C for 10 min. Spermatozoa were incubated in PBS containing 1% BSA, after which Alexa Fluor™ 594-phalloidin was added to PBS at a 1:100 dilution and incubated for 1 h.

To visualise nuclei, HCS NuclearMask (1:2000; cat. H10325, Thermo Fisher Inc.; excitation: 355 nm, emission: 460 nm) was added and incubated for a further 10 min. The NuclearMask marker was used solely to identify sperm nuclei and define the recognition area for analysis in ImageJ. Samples were air-dried at room temperature and mounted in glycerol. Spermatozoa were observed using a Carl Zeiss LSM 800 confocal microscope, and red fluorescence corresponding to polymerised actin in the sperm head was quantified. Fluorescence intensity was calculated using the formula: ΔF = (FS − FB)/FB, where FS is the fluorescence of the sample and FB is the background fluorescence.

### 2.8. Assessment of Ca^2+^_i_

Spermatozoa were thawed, washed, and maintained in non-capacitating TALP medium. Intracellular calcium levels (Ca^2+^_i_) were measured using the fluorescent indicator FLUO-3 AM (Thermo Fisher). Cells were incubated with the dye for 30 min, followed by centrifugation at 300× *g* for 5 min. After removing the supernatant, the pellets were resuspended in capacitating medium (with or without Ca^2+^) and transferred to a 96-well plate (50 µL per well). Fluorescence changes were recorded for 10 min in control and thapsigargin-treated cells (in media with or without Ca^2+^) using a Varioskan multimode reader (Thermo Fisher Inc.). Variations in fluorescence intensity were used as an index of intracellular Ca^2+^ fluctuations. As no ratiometric calibration was performed (no ionomycin/F_max or Mn^2+^ quench/F_min), data are presented as relative fluorescence changes (ΔF/F_0_) and provide semi-quantitative comparisons of cytosolic Ca^2+^ levels between experimental conditions. The difference between fluorescence at 5 min and the initial value (F_0_) was calculated, and twelve independent experimental replicates were analysed.

### 2.9. Complementary Eosin–Nigrosin Viability Assay

The eosin–nigrosin assay (eosin: Biopack Inc., C.I. 45380; nigrosin: C.I. 50420) was performed to assess sperm viability under incubation with thapsigargin, DMSO, or in the absence of the drug, following a protocol adapted from Agarwal et al. [28]. Aqueous solutions of 1% (*w*/*v*) Eosin Y and 10% (*w*/*v*) Nigrosin were prepared according to the manufacturers’ specifications (Eosin Y, CAS 17372-87-1; Nigrosin, CAS 8005-03-6; both from Biopack, Autónoma de Buenos Aires, Argentina). For each sample, 10 µL of sperm suspension was mixed with 20 µL of eosin and homogenised for approximately 15 s, after which 20 µL of nigrosin was added and mixed for a further 15 s. A thin smear was then prepared on a clean glass slide and allowed to air-dry at room temperature. The preparations were examined under a bright-field optical microscope at 100× magnification. Viable spermatozoa were identified by unstained heads, whereas non-viable spermatozoa exhibited pink-stained heads. Sperm viability was assessed by counting at least 175 spermatozoa per sample from 3 to 7 technical replicates. The percentage of viable spermatozoa was calculated as follows: (number of viable spermatozoa/total spermatozoa counted) × 100.

### 2.10. Statistical Analysis

Motility parameters were analysed using a mixed linear model (MixedLM, statsmodels library, Python 3.11.7) with false discovery rate (FDR) correction (see Appendix A). Pairwise significance markers indicated in the main figures refer to the specific post hoc comparisons described in the corresponding legends. Qualitative data were analysed using the χ^2^ test or Fisher’s exact test. Differences were considered statistically significant at *p* < 0.05. The legend of each figure specifies the statistical test applied to the corresponding dataset.

## 3. Results

### 3.1. SERCA Is Concentrated in the Acrosomal Region in Cryopreserved Bull Sperm

To analyse the relative abundance and subcellular localisation of SERCA in spermatozoa, cells were stained with a monoclonal anti-SERCA antibody. Images were examined to identify the fluorescent signal corresponding to SERCA. Positive SERCA labelling was observed in spermatozoa, with strong fluorescence detected in the head, particularly in the equatorial segment (Figure 1). Omission of the primary antibody resulted in no detectable fluorescence (Appendix A), confirming specificity. A merged image combining anti-SERCA and PSA staining was also included (Figure 1) to visualise the relative localisation of SERCA and the acrosome in spermatozoa.

### 3.2. SERCA Inhibition Affects the Motility of Cryopreserved Bull Sperm

The total motility of cryopreserved spermatozoa incubated in capacitation medium decreased over time. This reduction was exacerbated by the SERCA inhibitor thapsigargin (10 µM) added to the incubation medium at all analysed time points (Figure 2). We then assessed progressive motility, which reflects the ability of spermatozoa to move in a straight line or in a broad, curved trajectory with steady forward progression. Thapsigargin-treated samples at 20 and 30 min showed a significant reduction in the percentage of progressively motile spermatozoa compared with controls (Figure 2). Eosin–nigrosin staining revealed no significant differences in sperm viability among untreated (26.89 ± 3.26, *n* = 7), DMSO-treated (28.43 ± 3.11, *n* = 5), and thapsigargin-treated samples (24.42 ± 2.24, *n* = 3) (*p* > 0.05). Therefore, untreated sperm were used as the control group for all functional analyses (motility, capacitation, acrosome reaction, actin remodelling, and Ca^2+^_i_ handling).

Transient inhibition of SERCA initially increased the straight-line velocity (VSL) at 10 min, followed by a reduction at 20 min, and it reached values comparable to the control at 30 min (Appendix A). SERCA inhibition shows a tendency to reduce the curvilinear velocity (VCL) at 0 and 10 min and increase it at 30 min (Appendix A, Figure 3A). Average path velocity (VAP) was also elevated at 30 min (Figure 3B), as were beat-cross frequency (BCF) and amplitude of lateral head displacement (ALH) (Figure 3C,D), reaching significant differences in the last two parameters.

Straightness (STR), the ratio of VSL to VAP, which assesses the proportion of direct movement relative to the average, decreased at 20 and 30 min (Figure 3E, Appendix A). Thapsigargin induced a decrease in linearity (LIN) (Figure 3F), defined as the ratio of VSL to VCL, which reflects the straightness of the sperm trajectory.

Oscillation (WOB = VAP/VCL) is another parameter that reflects the regularity of sperm movement. WOB values, calculated as the ratio of average path velocity to curvilinear velocity, were significantly affected by thapsigargin (Appendix A).

Fractal dimension (FD) reflects the complexity of a trajectory and is used to assess the irregularity of sperm movement. Straight and regular trajectories correspond to low FD values, indicating more linear motion. Thapsigargin-treated spermatozoa exhibited higher FD values, suggesting curved and irregular trajectories with frequent changes in direction (Figure 3G).

Finally, we evaluated mean angular displacement (MAD), which quantifies changes in the direction of sperm movement over time. High MAD values indicate frequent directional changes, whereas lower values reflect more stable trajectories with minimal angular deviations. Spermatozoa treated with thapsigargin exhibited higher MAD (Figure 3H), suggesting increased frequency of directional changes. This behaviour is consistent with the onset of a hyperactivation-like motility pattern, characterised by vigorous, asymmetrical flagellar beating and frequent alterations in swimming direction.

Appendix A presents the values for all motility parameters across the time points described, while Appendix A shows a heatmap indicating significant changes. A representative video comparing control and thapsigargin-treated samples is provided in the Appendix A.

### 3.3. SERCA Plays a Key Role in the Progression of Capacitation in Bull Spermatozoa

To identify stages of the capacitation process, we analysed sperm acrosome integrity. Chlortetracycline (CTC) staining allows differentiation between cells with and without an acrosome, and further distinguishes capacitated from non-capacitated cells with intact acrosomes [29,30]. Non-capacitated spermatozoa exhibited fluorescence across the entire head and midpiece, whereas capacitated cells displayed fluorescence restricted to the acrosome. Spermatozoa that had undergone the acrosomal reaction showed no fluorescence or fluorescence in the acrosomal region oriented towards the centre of the head, referred to here as the equatorial line (Figure 4, left panel). Thapsigargin treatment consistently increased the proportion of spermatozoa undergoing the acrosomal reaction, with significant differences at 0, 10, and 30 min compared with controls (Figure 4, right panel). This effect was accompanied by a reduction in the number of capacitated cells, with both phenomena most pronounced at early time points (0 and 10 min).

### 3.4. SERCA Activity Modulates the Acrosomal Reaction

We also assessed whether SERCA influences the acrosomal reaction using PSA staining. PSA is a lectin that binds to mannose residues on the glycoproteins of the acrosomal outer membrane. When the acrosome is intact, fluorescence is observed in the acrosomal region, whereas it is absent following the acrosomal reaction [31]. Our experiments demonstrated that SERCA inhibition increased the proportion of spermatozoa undergoing the acrosomal reaction at 0, 10, and 20 min of incubation. Figure 5 presents representative images of spermatozoa with and without acrosomal reaction, while the accompanying graph depicts the proportion of spermatozoa that completed capacitation with an acrosomal reaction. As indicated by these results, thapsigargin may trigger the acrosomal reaction. However, since all experimental groups followed the same timeline, including exposure to caffeine and progesterone, basal capacitation and induced acrosomal reaction could not be fully distinguished under the present conditions.

### 3.5. SERCA Inhibition Alters the Timeline of Actin Polymerisation

Actin polymerisation occurs during sperm capacitation, and subsequent depolymerisation of F-actin to the globular form is required for the acrosomal reaction. In control spermatozoa, F-actin fluorescence was observed at 0 and 10 min, decreasing at 20 min of incubation in capacitation medium following progesterone-induced acrosomal reaction (Figure 6). Thapsigargin-treated spermatozoa exhibited a pronounced reduction in F-actin fluorescence at 10 min, consistent with actin depolymerisation preceding the acrosomal reaction. At 30 min, partial recovery of fluorescence was detected in both control and treated samples, suggesting reorganisation or stabilisation of actin filaments. These data represent relative changes in F-actin, as total actin content was not quantified.

### 3.6. Thapsigargin-Sensitive Ca^2+^ Sequestration Modulates Intracellular Ca^2+^ Levels in Cryopreserved Bull Spermatozoa

To investigate how SERCA activity regulates intracellular Ca^2+^ in cryopreserved bull spermatozoa, we measured changes in fluorescence using FLUO-3 AM, a cytosolic Ca^2+^ indicator. Spermatozoa were incubated with the dye, subdivided into control and thapsigargin-treated groups, and placed in a plate reader. Thapsigargin treatment resulted in increased fluorescence over time compared with controls, indicating elevated cytosolic Ca^2+^ levels. Figure 7 shows a representative trace of fluorescence changes over time. We then calculated the difference in fluorescence after 5 min relative to the initial value, revealing a significant increase in the presence of the SERCA inhibitor (Figure 7). To determine whether the observed increase in cytosolic Ca^2+^ originated from the extracellular medium, control and treated samples were incubated in TALP medium without Ca^2+^, supplemented with 5 mM EGTA. Under these conditions, fluorescence declined, and SERCA inhibition further accentuated this reduction (Figure 7).

## 4. Discussion

This study investigated the contribution of thapsigargin-sensitive Ca^2+^ sequestration, likely mediated by SERCA, to the regulation of intracellular Ca^2+^ homeostasis and functional parameters in cryopreserved bovine spermatozoa. Our data demonstrate that pharmacological inhibition of SERCA markedly affects motility patterns, capacitation status, actin dynamics, and the incidence of the acrosomal reaction. Although these findings highlight a significant role for SERCA in key physiological processes, they should be interpreted with caution and regarded as hypothesis-generating rather than providing definitive evidence of causal relationships.

Immunofluorescence analysis revealed predominant SERCA labelling in the acrosomal region, oriented towards the centre of the sperm head and concentrated along the equatorial line. This localisation differs from previous reports in several species, where SERCA has been observed throughout the acrosome or midpiece [10,12,13,24]. Such variation may reflect differences in antibody specificity, species-specific sperm architecture, or structural rearrangements induced by cryopreservation. However, while the observed fluorescence pattern supports localisation of SERCA within the acrosomal and equatorial regions in our study, further confirmation using simultaneous mitochondrial markers and additional negative controls (isotype- or peptide-blocked antibodies) would be required to exclude potential signal overlap and to strengthen organelle-level resolution.

Exposure to thapsigargin resulted in a marked reduction in progressive motility and an increase in hyperactivation-like parameters, including curvilinear velocity and lateral head displacement. These changes are consistent with elevated intracellular Ca^2+^ and resemble hyperactivation patterns described in other mammalian species [10,23]. The findings suggest that disruption of Ca^2+^ sequestration mechanisms promotes premature transitions in motility, which may shorten the functional lifespan of thawed spermatozoa. One limitation in motility measures is that although CASA systems are often used in research and regular practices, CASA motility parameters are obtained by analysing 2D trajectories of motion using videos of sperm movement. Recent studies proposed evaluating 3D sperm movements (helical movements with 3D rotation and 3D full-type hyperactivation) for a more precise analysis of sperm motility [32].

Thapsigargin treatment also increased the proportion of acrosome-reacted spermatozoa and reduced the fraction classified as capacitated. These effects are consistent with the hypothesis that elevated intracellular Ca^2+^ can trigger premature acrosomal exocytosis, particularly in cryopreserved samples that already exhibit membrane alterations and partial capacitation (cryo-capacitation) [33]. However, as fresh sperm were not evaluated, the assertion that SERCA “remains functional” following cryopreservation cannot be directly substantiated. Rather, our findings indicate that a thapsigargin-sensitive component of the Ca^2+^ homeostatic machinery remains responsive and influences downstream processes in thawed spermatozoa.

The disruption of actin polymerisation observed following SERCA inhibition supports the notion that elevated Ca^2+^_i_ promotes F-actin depolymerisation during the acrosomal reaction. This remodelling likely reflects an indirect consequence of Ca^2+^ dysregulation rather than a direct effect of SERCA on the cytoskeleton. Furthermore, the increase in intracellular Ca^2+^ was abolished under Ca^2+^-free conditions, suggesting that store depletion triggers extracellular Ca^2+^ influx, consistent with store-operated Ca^2+^ entry described in several mammalian sperm models [34]. Although the use of FLUO-3 AM without ratiometric calibration does not permit quantification of absolute intracellular Ca^2+^ concentrations, this method is widely employed to assess relative changes in Ca^2+^ dynamics in spermatozoa [24,35]. Therefore, the increase in fluorescence observed after SERCA inhibition should be interpreted as a relative rise in cytosolic Ca^2+^, reflecting blocked Ca^2+^ reuptake into intracellular stores rather than activation of a specific signalling pathway.

Nevertheless, the potential contribution of other Ca^2+^ transport systems—particularly the secretory pathway Ca^2+^-ATPase (SPCA) and the plasma membrane Ca^2+^-ATPase (PMCA)—should be considered. SPCA plays a significant role in spermatozoa [10,36] and may complement or compensate for SERCA activity under cryopreserved conditions. Elucidating the relative contributions of these pumps will require future studies employing both pharmacological and molecular approaches.

Altogether, the results indicate that thapsigargin-sensitive Ca^2+^ sequestration mechanisms contribute to the regulation of motility, actin organisation, and acrosomal stability in cryopreserved bovine spermatozoa. These mechanisms are likely involved in the premature capacitation and reduced fertilising potential commonly observed following freezing and thawing. A better understanding of how SERCA, SPCA, PMCA, and CatSper channels coordinate Ca^2+^ homeostasis in mammalian sperm [7,8,9] may provide novel strategies to mitigate cryo-induced dysfunction and improve post-thaw fertility.

Although thapsigargin is a well-established SERCA inhibitor, its effects on intracellular Ca^2+^ may also be accompanied by endoplasmic reticulum stress and the generation of reactive oxygen species (ROS), particularly in somatic cells exposed for prolonged periods [37,38,39]. In spermatozoa, however, short-term incubation with thapsigargin primarily induces Ca^2+^ mobilisation without clear evidence of oxidative damage [40,41]. In the present study, sperm were exposed to thapsigargin for limited periods under controlled conditions, suggesting that the observed effects on (Ca^2+^)_i_ and motility parameters reflect direct perturbation of Ca^2+^ handling rather than generalised cellular stress. Nevertheless, the potential contribution of ROS cannot be excluded, and future experiments assessing oxidative status in the presence of thapsigargin would be valuable to confirm this interpretation.

One limitation of this study is the use of semen from a single bull and the absence of fresh-sperm controls. Consequently, our conclusions are confined to mechanistic interpretations under cryopreserved conditions. Moreover, the 10 µM thapsigargin concentration used here is substantially higher than commonly reported nanomolar doses for selective SERCA inhibition [36], and therefore off-target and cytotoxic components cannot be excluded. Future studies should incorporate multiple biological replicates, a range of thapsigargin concentrations to minimise potential off-target effects, and alternative inhibitors such as CPA or BHQ to confirm specificity.

Taken together, the present results—obtained from a single bull under specific cryopreservation and incubation conditions—support the hypothesis that SERCA, a thapsigargin-sensitive Ca^2+^ pump, contributes to the regulation of intracellular Ca^2+^ dynamics, motility, and acrosomal events (Figure 8). Rather than indicating exclusively preserved SERCA function, the findings highlight that components of the Ca^2+^-regulatory network remain responsive to pharmacological perturbation following cryopreservation. This insight advances our understanding of post-thaw sperm physiology and may inform future strategies to enhance fertilisation efficiency in cattle breeding programmes. However, broader validation across individuals and preservation protocols will be necessary to generalise these observations.

## Figures and Tables

**Figure 1 cells-14-01892-f001:**
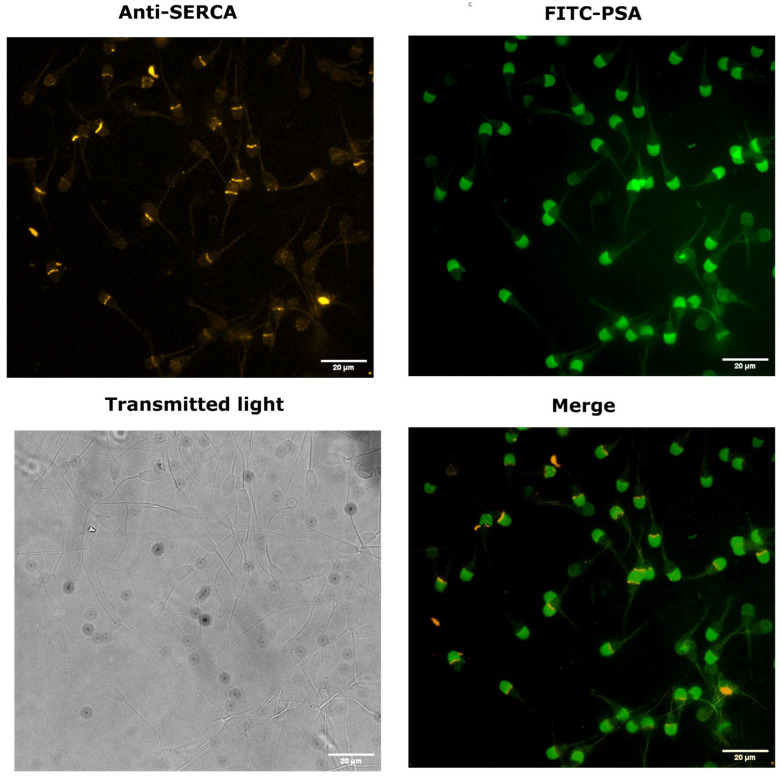
Representative images showing the localisation of SERCA in cryopreserved bull spermatozoa. (**Left, upper panel**): spermatozoa stained with a monoclonal anti-SERCA antibody followed by a Cy3-conjugated secondary antibody. (**Left, lower panel**): the corresponding field visualised under transmitted light. (**Right, upper panel**): acrosomal region revealed by FITC–PSA staining. SERCA is predominantly localised in the acrosomal area, oriented towards the centre of the sperm head (equatorial segment), as shown in the merged image (**Right, lower panel**). *n* = 307 spermatozoa from three independent experiments.

**Figure 2 cells-14-01892-f002:**
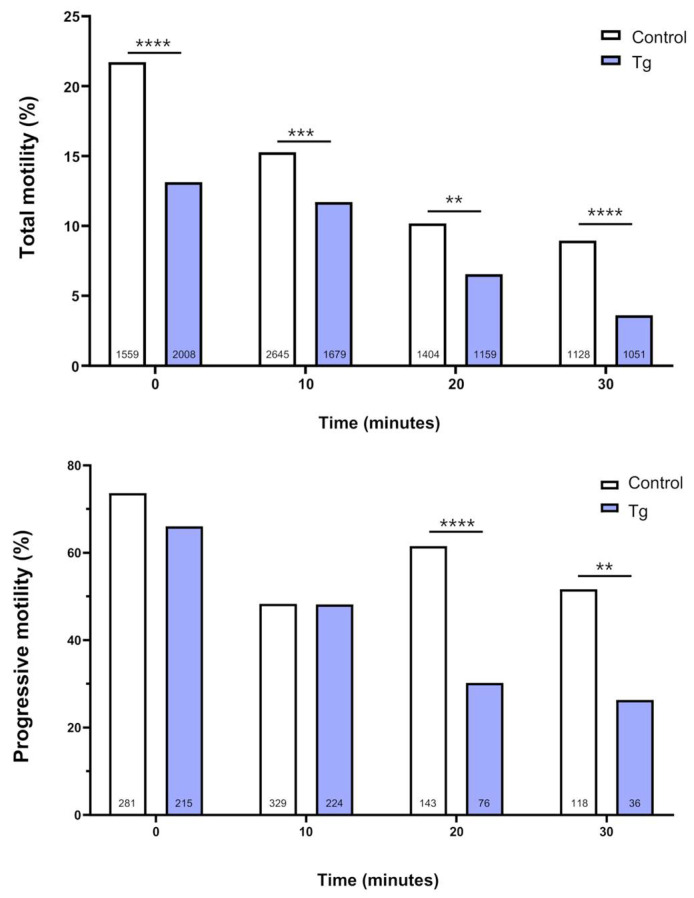
Effects of SERCA inhibition on sperm motility. (**Upper graph**) Percentage of motile spermatozoa at 0, 10, 20, and 30 min in control samples and in samples treated with 10 µM thapsigargin (Tg) in capacitation medium. (**Bottom graph**) Percentage of progressively motile spermatozoa at the same time points. Four independent experiments were performed. The total number of cells analysed for each group is indicated at the base of the bars in both graphs. Comparisons between control and Tg-treated cells were performed using Fisher’s exact test at each time point. ** *p* < 0.01; *** *p* < 0.001; **** *p* < 0.0001.

**Figure 3 cells-14-01892-f003:**
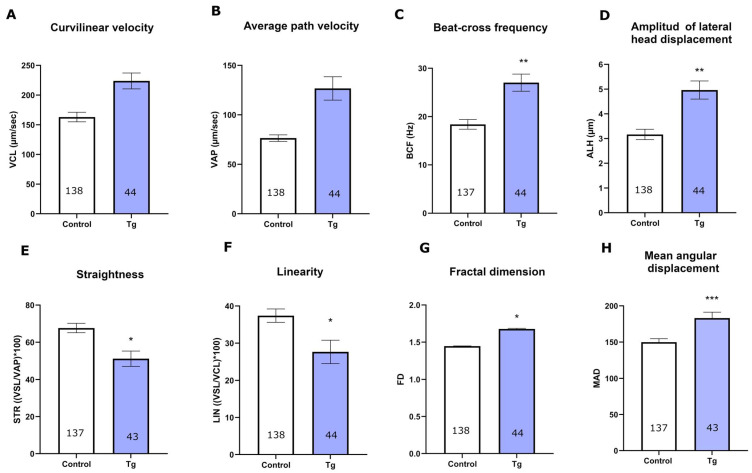
Analysis of motility parameters in control and thapsigargin-treated spermatozoa. Graphs depict various motility parameters measured at 30 min in cells incubated in capacitation medium following the addition of progesterone to induce the acrosomal reaction. Each parameter compares the control and Tg-treated groups. VCL: curvilinear velocity; VAP: average path velocity; BCF: beat-cross frequency; ALH: amplitude of lateral head displacement; STR: straightness; LIN: linearity; DNC: dance; FD: fractal dimension; MAD: mean angular displacement. All parameters indicate that SERCA inhibition enhanced hyperactivation. Motility parameters were analysed using a mixed linear model with false discovery rate (FDR) correction. Pairwise significance markers refer to the specific post hoc comparisons. The total number of cells from four independent experiments is indicated at the base of the bars. (**A**) VCL: curvilinear velocity; (**B**) VAP: average path velocity; (**C**) BCF: beat-cross frequency; (**D**) ALH: amplitude of lateral head displacement; (**E**) STR: straightness; (**F**) LIN: linearity; DNC: dance; (**G**) FD: fractal dimension; (**H**) MAD: mean angular displacement. * *p* < 0.05; ** *p* < 0.01; *** *p* < 0.001.

**Figure 4 cells-14-01892-f004:**
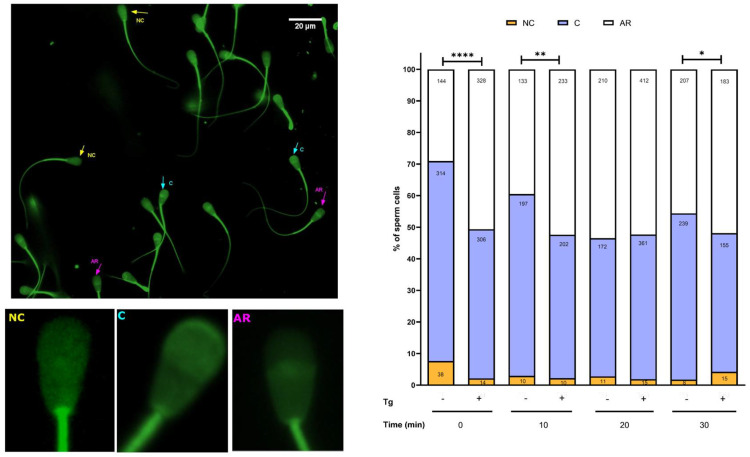
Stages of sperm capacitation with and without SERCA inhibition. (**Left**): Representative image showing a group of spermatozoa with different fluorescence patterns. Arrows indicate: NC, non-capacitated (yellow); C, capacitated (cyan); RA, acrosome-reacted (purple). Insets at the bottom show individual sperm heads highlighting the fluorescence patterns. (**Right**): Comparison of the percentage of non-capacitated, capacitated, and acrosome-reacted spermatozoa at different time points in control and thapsigargin-treated samples. A chi-square test was performed to evaluate differences between control and treated cells at each time point. Bars represent the mean proportion of cells at each stage from four independent experiments. The total number of cells analysed for each condition and time point is indicated on the bars. Fisher’s exact test was also applied to assess differences between groups. Significant differences were observed at 0, 10, and 20 min. * *p* < 0.05; ** *p* < 0.01; **** *p* < 0.0001.

**Figure 5 cells-14-01892-f005:**
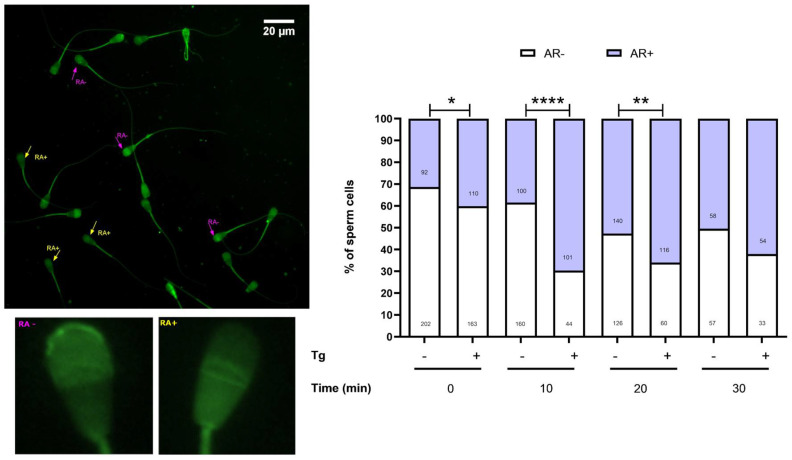
Effect of SERCA inhibition on the acrosomal reaction. (**Left**): Representative image of spermatozoa exhibiting different fluorescence patterns. Arrows indicate RA−, spermatozoa without acrosomal reaction, and RA+, spermatozoa with acrosomal reaction. Insets at the bottom show individual sperm heads highlighting the fluorescence patterns. (**Right**): Comparison of the percentage of RA− and RA+ spermatozoa at different time points in control and thapsigargin-treated samples. Bars represent the mean proportion of cells in each category from three independent experiments. Fisher’s exact test was used to evaluate differences between groups, and the associations were statistically significant. The total number of cells analysed for each condition and time point is indicated on the bars. * *p* < 0.05; ** *p* < 0.01; **** *p* < 0.0001.

**Figure 6 cells-14-01892-f006:**
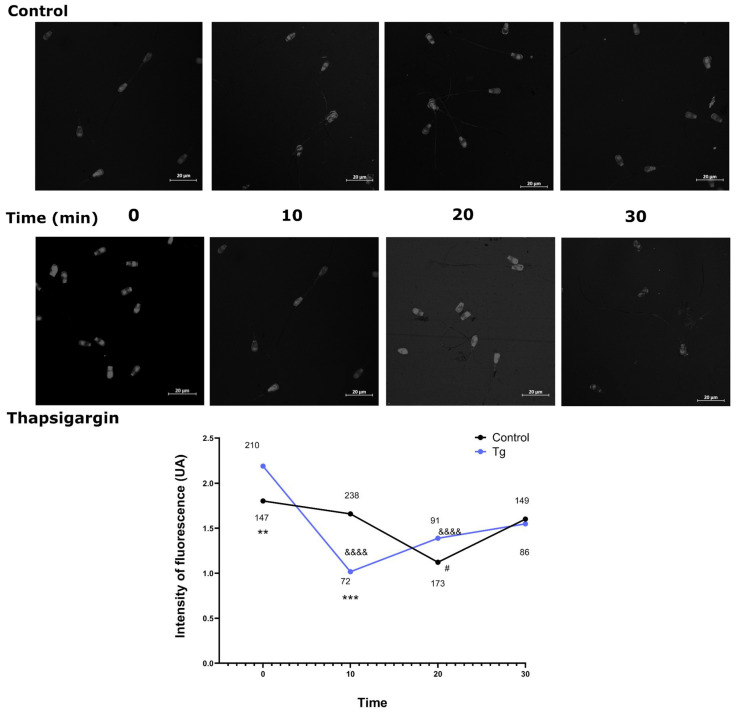
Effect of SERCA inhibition on actin polymerisation. (**Top**): Representative images of spermatozoa stained with Alexa Fluor™ 594-phalloidin, showing F-actin in control and thapsigargin-treated groups at 0, 10, 20, and 30 min. (**Bottom**): Quantification of actin polymerisation over time in cryopreserved bull spermatozoa in the presence (blue circles) or absence (black circles) of thapsigargin (10 µM). Values adjacent to the symbols indicate the number of cells analysed. Data were obtained from four independent experiments using four different straws from the same batch of cryopreserved sperm. Comparisons between control and thapsigargin-treated groups at each time point were performed using the Mann–Whitney test (*). Comparisons across all time points within each group were performed using the Kruskal–Wallis test (# for control; & for thapsigargin). For all analyses, *p* < 0.05 was considered statistically significant. Symbols indicate significance levels: #, *p* < 0.05; **, *p* < 0.01; ***, *p* < 0.001; &&&& *p* < 0.0001.

**Figure 7 cells-14-01892-f007:**
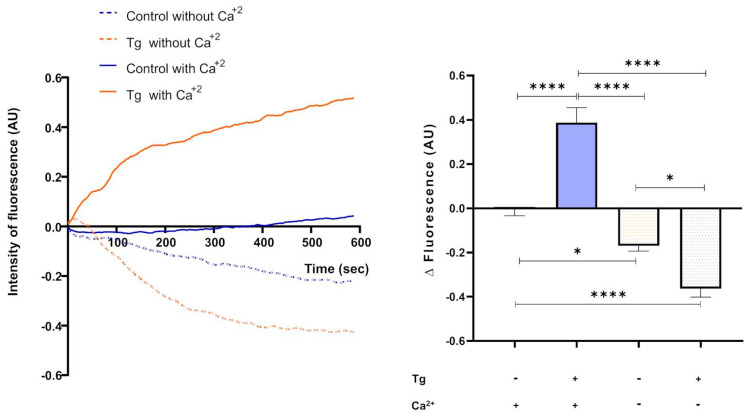
Intracellular Ca^2+^ signalling in the presence and absence of SERCA inhibition. (**Left**) Time course of FLUO-3 AM fluorescence in four groups of spermatozoa: control and thapsigargin-treated (Tg), each in the presence or absence of extracellular Ca^2+^. (**Right**) Mean ± SEM from two technical replicates of four independent experiments. Thapsigargin-treated spermatozoa exhibited a significant increase in fluorescence in medium containing Ca^2+^ compared with controls, whereas samples incubated in Ca^2+^-free medium showed a reduction in fluorescence. Comparisons were performed using two-way ANOVA. * *p* < 0.05; **** *p* < 0.0001.

**Figure 8 cells-14-01892-f008:**
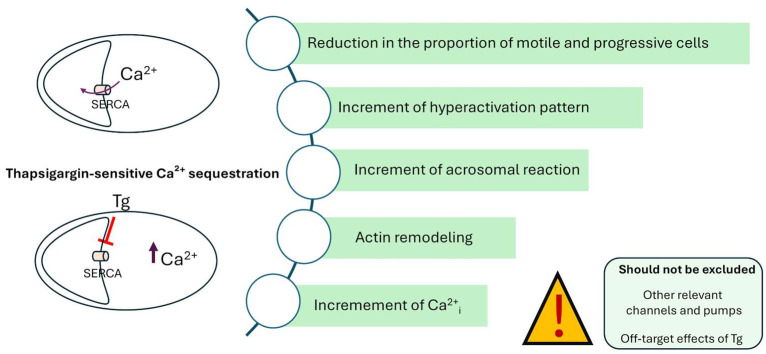
Summary of the main findings and proposed mechanism. Schematic representation of the effects of thapsigargin (Tg) on Ca^2+^ handling and functional parameters in cryopreserved bovine spermatozoa. Under normal conditions, SERCA contributes to Ca^2+^ sequestration within intracellular stores. Inhibition of SERCA by Tg leads to an increase in cytosolic Ca^2+^ concentration (↑ Ca^2+^_i_), associated with reduced motility and progressive movement, increased hyperactivation, enhanced acrosomal reaction, and actin remodelling. The diagram also highlights potential confounding factors that cannot be excluded, including the involvement of other Ca^2+^ channels and pumps, as well as possible off-target effects of Tg.

## Data Availability

The datasets used and/or analysed during the current study are available from the corresponding author on reasonable request.

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
