# Peer review of "Role of the Ca^2+^-ATPase Pump (SERCA) in Capacitation and the Acrosome Reaction of Cryopreserved Bull Spermatozoa"

_cells, 2025, doi:10.3390/cells14231892_

Round 1
Reviewer 1 Report
Comments and Suggestions for Authors
Comments for the Authors
This manuscript addresses the potential role of the sarco/endoplasmic reticulum Ca²⁺-ATPase (SERCA) in the regulation of capacitation and the acrosome reaction of cryopreserved bull spermatozoa. Although the topic is relevant for reproductive physiology and bovine assisted reproduction, the current version suffers from major methodological flaws, conceptual inconsistencies, and over-interpretation of data that seriously compromise the validity of its conclusions.
Below I provide detailed, section-by-section comments.
General and conceptual issues
- Non-specific pharmacological design.
SERCA was inhibited with 10 µM thapsigargin (Tg)—a concentration ~1000-fold higher than the nanomolar range used for selective inhibition. At 10 µM, Tg is cytotoxic, disrupts plasma and acrosomal membranes, and alters Ca²⁺ homeostasis by multiple off-target effects. Therefore, the observed reductions in motility and premature acrosome reactions cannot be attributed specifically to SERCA inhibition. A dose–response curve (1–300 nM + 1–3 µM) and at least one alternative SERCA inhibitor (e.g., CPA or BHQ) are required to support causal interpretation.
- Lack of biological replication (pseudoreplication).
All samples derive from a single bull and ejaculate, with “straws from the same batch” treated as independent replicates. These are technical, not biological, replicates; thus, statistical independence is violated. Conclusions about “bull sperm” cannot be generalized. Future experiments must include ≥ 3 bulls (biological replicates).
- Missing essential controls.
- No vehicle control (DMSO) for Tg.
- No viability assays (PI/Hoechst or eosin–nigrosin) after Tg exposure.
- No groups without caffeine or without progesterone to dissociate basal capacitation from induced acrosome reaction.
- No co-localization of SERCA with acrosomal or mitochondrial markers, nor negative (isotype or peptide-blocked) controls.
- Statistical design.
Numerous variables (CASA parameters × times × treatments) were compared with multiple t-tests and ANOVAs without any correction for multiple testing. The lack of mixed-model structure (treatment × time as fixed factors; “experiment” or “batch” as random) further inflates type-I error. Re-analysis with linear mixed models and FDR correction is needed.
- Terminology and internal inconsistencies.
- TALP “non-capacitating” medium contains 20 mM NaHCO₃, which makes it capacitating.
- PSA-FITC concentration stated as 50 mg/mL is unfeasible; likely 50 µg/mL.
- Fluorophores are inconsistent (Methods: Alexa 594-phalloidin; Fig. 7: TRITC-phalloidin).
- Abbreviations vary (e.g., “FD/FC”, “WOB/WOS”, “SLV/VSL”).
- Antibody validation for SERCA is absent.
- Over-stated conclusions.
The claim that “SERCA accelerates events leading to the acrosome reaction and remains functional after cryopreservation” is not experimentally supported. No comparison with fresh semen was made; thus, “not altered by cryopreservation” cannot be concluded.
Section-specific remarks
Title and Abstract
- Use correct capitalization: “Role of the Ca²⁺-ATPase pump (SERCA) in capacitation and the acrosome reaction of cryopreserved bull spermatozoa.”
- The abstract asserts direct responsibility of SERCA for Ca²⁺ reuptake into the acrosome, which remains hypothetical here. Revise to a conditional phrasing (“may contribute to”).
- Limit mechanistic claims and acknowledge methodological constraints (single bull, pharmacological specificity).
Introduction
- The overview of capacitation is accurate but oversimplifies Ca²⁺ regulation, ignoring SPCA, PMCA, and CatSper channels.
- Lacks a clear, testable hypothesis. Suggest: “We hypothesized that inhibition of SERCA would perturb Ca²⁺ handling and alter capacitation dynamics in cryopreserved bull spermatozoa.”
- The phenomenon of cryocapacitation should be presented as a confounding variable, not merely background.
Materials and Methods
- The “non-capacitating” TALP medium includes bicarbonate and BSA—both capacitating agents. Revise composition.
- Explicitly state that all experiments used straws from one bull/ejaculate.
- The temporal scheme (Fig. 1) is unclear: Tg was added for 2 min at all time points, meaning t = 0 already includes Tg exposure. Provide an absolute timeline diagram.
- Missing vehicle control, viability control, and details of CASA video acquisition (fps, temperature, number of tracks analyzed).
- FLUO-3 AM fluorescence was recorded in a plate reader without ratiometric calibration (no ionomycin/Fmax, no Mn²⁺ quench); therefore, results are semi-quantitative.
- Immunofluorescence: co-staining with PSA/PNA and MitoTracker is essential to confirm localization.
Results
- SERCA localization (Fig. 2): fluorescence is strongest in the equatorial/post-acrosomal region, not the acrosome; the claim of “reuptake into the acrosome” is inconsistent with the data.
- Motility (Figs. 3–4): decreases in total/progressive motility with Tg likely reflect cytotoxicity, not physiological modulation. “Hyperactivation” inferred from increased ALH/VCL/BCF is speculative without 3D-tracking or energy expenditure data.
- Capacitation (Fig. 5) and Acrosome Reaction (Fig. 6): experimental timeline confounds interpretation; CTC and PSA concentrations inconsistent; replicate number too small (n = 3–4).
- Actin polymerization (Fig. 7): the interpretation of “repolymerization at 30 min” lacks mechanistic support; no total actin quantification.
- Ca²⁺ assay (Fig. 8): absence of calibration precludes quantitative comparison; fluorescence increase after Tg merely reflects blocked reuptake, not physiological signaling.
Discussion
- Over-extends conclusions from phenomenological correlations.
- Contradiction: localization reported in the equatorial segment but discussed as acrosomal.
- The statement that SERCA “remains functional after cryopreservation” is speculative; fresh sperm were never tested.
- The possibility of SPCA involvement is acknowledged nowhere; this omission weakens mechanistic depth.
- Many paragraphs reiterate prior literature rather than critically integrate the present data.
- Rewrite with a cautious tone (“our findings suggest…”, “consistent with the hypothesis that…”).
References
- Generally appropriate, but some typographical errors (“Journal of Androloy”) and inconsistent formatting.
- Missing key references on SPCA, PMCA, and CatSper in bovine sperm.
The manuscript is generally understandable, but numerous grammatical and syntactic errors compromise fluency and precision. The text alternates between American and British English, with inconsistent capitalization of technical terms (e.g., Atpase vs. ATPase, acrosomal reaction vs. acrosome reaction). Sentence structure is sometimes verbose, and transitions between sections lack coherence.
A thorough professional English revision is required to correct terminology, improve readability, and ensure stylistic consistency throughout the text.
Author Response
Reviewer 1
Comments and Suggestions for Authors
Comments for the Authors
This manuscript addresses the potential role of the sarco/endoplasmic reticulum Ca²⁺-ATPase (SERCA) in the regulation of capacitation and the acrosome reaction of cryopreserved bull spermatozoa. Although the topic is relevant for reproductive physiology and bovine assisted reproduction, the current version suffers from major methodological flaws, conceptual inconsistencies, and over-interpretation of data that seriously compromise the validity of its conclusions.
Below I provide detailed, section-by-section comments.
General and conceptual issues
- Non-specific pharmacological design.
SERCA was inhibited with 10 µM thapsigargin (Tg)—a concentration ~1000-fold higher than the nanomolar range used for selective inhibition. At 10 µM, Tg is cytotoxic, disrupts plasma and acrosomal membranes, and alters Ca²⁺ homeostasis by multiple off-target effects. Therefore, the observed reductions in motility and premature acrosome reactions cannot be attributed specifically to SERCA inhibition. A dose–response curve (1–300 nM + 1–3 µM) and at least one alternative SERCA inhibitor (e.g., CPA or BHQ) are required to support causal interpretation
The dose selection was based on the analysis by Ho and Suarez on bovine sperm (Ho and Suarez, 2001), Williams and Ford (Williams and Ford, 2003), and Durithala et al. (2022). In Ho and Suarez (2001), the authors performed a dose–response curve. Based on their results, we selected 10 μM, a moderate dose that is sufficiently effective to evaluate changes in motility and the acrosomal reaction, while allowing a reasonably short-term incubation to minimise potential off-target effects. In studies with other Ca²⁺ pumps, such as SERCA1A, CPA appears to be a less potent inhibitor (Chen et al., 2017). While CPA also increases intracellular Ca²⁺ and inhibits motility, its effects are generally less pronounced, requiring higher concentrations to achieve a similar outcome. BHQ, on the other hand, is the least potent and specific of the three, with possible off-target effects on other membrane proteins.
We have revised the manuscript to clarify that the effects observed at this concentration may include off-target or cytotoxic components, and we now present our conclusions as mechanistic rather than exclusively SERCA-specific. Future work could include a full dose–response analysis and alternative SERCA inhibitors (CPA, BHQ) to strengthen the causal interpretation. Please see the Discussion section, page 22, lines 363–365.
References
Ho HC, Suarez SS. An inositol 1,4,5-trisphosphate receptor-gated intracellular Ca(2+) store is involved in regulating sperm hyperactivated motility. Biol Reprod. 2001 Nov;65(5):1606-15. doi: 10.1095/biolreprod65.5.1606. PMID: 11673282.
Williams KM, Ford WC. Effects of Ca-ATPase inhibitors on the intracellular calcium activity and motility of human spermatozoa. Int J Androl. 2003 Dec;26(6):366-75. doi: 10.1111/j.1365-2605.2003.00438.x. PMID: 14636222.
Duritahala, Sakase M, Harayama H. Involvement of Ca2+-ATPase in suppressing the appearance of bovine helically motile spermatozoa with intense force prior to cryopreservation. J Reprod Dev. 2022 Jun 1;68(3):181-189. doi: 10.1262/jrd.2021-143. Epub 2022 Mar 3. PMID: 35236801; PMCID: PMC9184823.
Chen J, De Raeymaecker J, Hovgaard JB, Smaardijk S, Vandecaetsbeek I, Wuytack F, Møller JV, Eggermont J, De Maeyer M, Christensen SB, Vangheluwe P. Structure/activity relationship of thapsigargin inhibition on the purified Golgi/secretory pathway Ca2+/Mn2+-transport ATPase (SPCA1a). J Biol Chem. 2017 Apr 28;292(17):6938-6951. doi: 10.1074/jbc.M117.778431. Epub 2017 Mar 6. PMID: 28264934; PMCID: PMC5409463.
- Lack of biological replication (pseudoreplication).
All samples derive from a single bull and ejaculate, with “straws from the same batch” treated as independent replicates. These are technical, not biological, replicates; thus, statistical independence is violated. Conclusions about “bull sperm” cannot be generalized. Future experiments must include ≥ 3 bulls (biological replicates).
We acknowledge the reviewer’s point regarding biological replication. All samples were obtained from frozen straws derived from a single bull and ejaculate; therefore, the data represent technical rather than biological replicates. This design was intentionally chosen to minimise inter-individual variability and to establish a controlled, mechanistic proof-of-concept of thapsigargin’s effects on Ca²⁺ homeostasis and sperm function. We have clarified this limitation in the revised manuscript and noted that future studies should include biological replicates to validate and generalise the findings. Please see the Discussion section, page 22, lines 361–363.
- Missing essential controls.
- No vehicle control (DMSO) for Tg.
We appreciate the reviewer’s observation. To ensure that the vehicle did not affect sperm physiology, we performed viability assays (eosin–nigrosin) comparing untreated, DMSO-treated, and Tg-treated samples, as suggested. No significant differences in cell viability were detected among these groups (p > 0.05). Based on these results, and considering that sperm motility is strongly dependent on viability, we did not include an additional DMSO control in the motility assays. This information has now been incorporated into the Materials and Methods and Results sections. Please see pages 8-9, lines 173–186 in the Methods section and pages 10-11, lines 210–213 in the Results section.
- No viability assays (PI/Hoechst or eosin–nigrosin) after Tg exposure.
Thanks to the reviewer for the suggestion. We carried out the eosin-nigrosin assay to evaluate the viability of sperm cells under incubation with thapsigargin, DSMO, and in the absence of drug. No significant differences in the proportion of viable cells were detected among the groups (p > 0.05). We added a paragraph in the material and methods sections, results, and a supplementary figure. Please, see pages 8-9, lines 173–186 in the methods section and pages 10-11, lines 210–213 in the results section.
- No groups without caffeine or without progesterone to dissociate basal capacitation from induced acrosome reaction.
Separating basal from induced responses would indeed provide finer resolution. In our experimental design, caffeine and progesterone were applied sequentially to mimic physiological capacitation and acrosome induction. As noted in the Results section, basal capacitation and induced acrosome reaction could not be fully dissociated under these conditions. Please see page 15, lines 267–269.
- No co-localization of SERCA with acrosomal or mitochondrial markers, nor negative (isotype or peptide-blocked) controls.
The monoclonal antibody (Santa Cruz Biotechnology, sc-8095, clone F-1) was generated using a peptide derived from the N-terminus of human SERCA2, and it has been tested by the manufacturer (see manufacturer's datasheet: thttps://www.scbt.com/es/p/serca2-antibody-f-1).
We have confirmed through bioinformatic analysis that this specific immunogen region (residues 2-29) is 100% identical between the human (UniProt: ATP2A2_HUMAN (UniProt accession P16615) and bovine (UniProt accession A0AAA9T933) orthologs. This sequence conservation provides a strong theoretical basis for the expected cross-reactivity. Furthermore, the antibody has been validated by the manufacturer (see manufacturer's datasheet: https://www.scbt.com/es/p/serca2-antibody-f-1). Our experimental results provide direct evidence of its specificity. Importantly, our negative control (omitting the primary antibody) showed no signal, confirming the specificity of the observed fluorescence. See supplementary figure 1. Moreover, we included a merged image combining the anti-SERCA antibody and PSA staining to visualise both SERCA and the acrosome in spermatozoa. This control has been added to Figure 1 in the Results section (page 10), to enhance clarity.
UniProt. (2025, October). ATP2A2_HUMAN (UniProt accession P16615) and ATP2A2_BOVIN (UniProt accession A0AAA9T933). UniProtKB protein knowledgebase. Retrieved October 16, 2025, from https://www.uniprot.org
- Statistical design.
Numerous variables (CASA parameters × times × treatments) were compared with multiple t-tests and ANOVAs without any correction for multiple testing. The lack of mixed-model structure (treatment × time as fixed factors; “experiment” or “batch” as random) further inflates type-I error. Re-analysis with linear mixed models and FDR correction is needed.
We thank the reviewer for this comment. In the revised manuscript, we carried out the analysis using a mixed linear model (MixedLM, statsmodels library, Python) with false discovery rate (FDR) correction. Please see page 9, lines 188-191, supplementary table 1, and supplementary figure 2 showing the heatmap.
- Terminology and internal inconsistencies.
- TALP “non-capacitating” medium contains 20 mM NaHCO₃, which makes it capacitating.
We appreciate the reviewer’s observation regarding the presence of NaHCO₃ in the TALP “non-capacitating” medium. Bicarbonate is known to activate the sAC–cAMP–PKA pathway and induce membrane changes associated with sperm capacitation (Soriano-Úbeda, 2019). Although bovine sperm can respond to heparin and other capacitation stimuli (Parrish, 1988), bicarbonate alone—without albumin/cholesterol acceptors or optimal Ca²⁺ concentrations—is insufficient to induce full capacitation over short incubation periods (Soriano-Úbeda, 2019). Therefore, while bicarbonate may trigger some early capacitation-related events, the medium employed here is considered non-capacitating in functional terms, as it does not support the complete sequence leading to capacitation or the acrosome reaction.
Parrish JJ, Susko-Parrish J, Winer MA, First NL. Capacitation of bovine sperm by heparin. Biol Reprod. 1988 Jun;38(5):1171-80. doi: 10.1095/biolreprod38.5.1171. PMID: 3408784.
Soriano-Úbeda C, Romero-Aguirregomezcorta J, Matás C, Visconti PE, García-Vázquez FA. Manipulation of bicarbonate concentration in sperm capacitation media improves in vitro fertilisation output in porcine species. J Anim Sci Biotechnol. 2019 Mar 11;10:19. doi: 10.1186/s40104-019-0324-y. PMID: 30899459; PMCID: PMC6410524.
- PSA-FITC concentration stated as 50 mg/mL is unfeasible; likely 50 µg/mL.
Thank you for noticing this typographical error. The correct concentration is 50 µg/mL. Please, see page 7, line 142.
- Fluorophores are inconsistent (Methods: Alexa 594-phalloidin; Fig. 7: TRITC-phalloidin).
Thank you for the observation. We acknowledge this inconsistency. Alexa Fluor™ 594-phalloidin was used in all experiments. The mention of TRITC-phalloidin in Figure 7 was a labeling error. Please refer to Figure 6 in the revised manuscript for the corrected information (legend, pages 17-18).
- Abbreviations vary (e.g., “FD/FC”, “WOB/WOS”, “SLV/VSL”).
We agree and have standardized all abbreviations throughout the manuscript to ensure consistency: FD, WOB, VSL. Please, see page 24.
Antibody validation for SERCA is absent.
We thank the reviewer for raising this point regarding antibody specificity. The monoclonal antibody (Santa Cruz Biotechnology, sc-8095, clone F-1) was generated using a peptide derived from the N-terminus of human SERCA2. Bioinformatic analysis confirmed that this immunogen region (residues 2–29) is 100% identical between human (UniProt: ATP2A2_HUMAN, P16615) and bovine (UniProt: A0AAA9T933) orthologues, providing a strong theoretical basis for expected cross-reactivity. Furthermore, the antibody has been validated by the manufacturer (datasheet: https://www.scbt.com/es/p/serca2-antibody-f-1). Our experimental results support its specificity: the negative control, omitting the primary antibody, produced no signal. This control has been incorporated in Supplementary Figure 1 for clarity.
UniProt. (2025, October). ATP2A2_HUMAN (UniProt accession P16615) and ATP2A2_BOVIN (UniProt accession A0AAA9T933). UniProtKB protein knowledgebase. Retrieved October 16, 2025, from https://www.uniprot.org
- Over-stated conclusions.
The claim that “SERCA accelerates events leading to the acrosome reaction and remains functional after cryopreservation” is not experimentally supported. No comparison with fresh semen was made; thus, “not altered by cryopreservation” cannot be concluded.
We appreciate this point. We rewrote the discussion section. Please, see pages 19-22, including modifications suggested by all reviewers.
Section-specific remarks
Title and Abstract
- Use correct capitalization: “Role of the Ca²⁺-ATPase pump (SERCA) in capacitation and the acrosome reaction of cryopreserved bull spermatozoa.”
Thank you, we have made the change. Please, see page 1, lines 1-2.
- The abstract asserts direct responsibility of SERCA for Ca²⁺ reuptake into the acrosome, which remains hypothetical here. Revise to a conditional phrasing (“may contribute to”).
We made the change. Please, see page 2, line 32.
- Limit mechanistic claims and acknowledge methodological constraints (single bull, pharmacological specificity).
We answered above, with modifications included in the new version.
Introduction
- The overview of capacitation is accurate but oversimplifies Ca²⁺ regulation, ignoring SPCA, PMCA, and CatSper channels.
We agree and will expand the Introduction and Discussion to incorporate SPCA, PMCA, and CatSper, with appropriate citations. Page 3, lines 49-50.
- Lacks a clear, testable hypothesis. Suggest: “We hypothesized that inhibition of SERCA would perturb Ca²⁺ handling and alter capacitation dynamics in cryopreserved bull spermatozoa.”
We have added a concise hypothesis: “We hypothesised that inhibition of SERCA would disrupt Ca²⁺ regulation and modify capacitation dynamics in cryopreserved bovine spermatozoa”. Page 4, lines 73-74.
- The phenomenon of cryocapacitation should be presented as a confounding variable, not merely background.
This variable has been highlighted in the introduction section as …Cryopreservation is known to induce capacitation-like membrane and signaling changes in spermatozoa, a phenomenon termed cryocapacitation. This process may partially activate sperm before experimental incubation, thereby influencing subsequent responses to capacitating agents or pharmacological treatments…” Pages 4, lines 70-72.
Materials and Methods
- The “non-capacitating” TALP medium includes bicarbonate and BSA—both capacitating agents. Revise composition.
We appreciate the reviewer’s observation regarding the presence of NaHCO₃ in the TALP “non-capacitating” medium. Bicarbonate is known to activate the sAC–cAMP–PKA pathway and elicit membrane changes associated with sperm capacitation (Soriano-Úbeda 2019) and although bovine sperm can respond to heparin and other capacitation stimuli (Parrish 1988), bicarbonate alone (in the absence of albumin/cholesterol acceptors or optimal Ca²⁺) is insufficient to induce full capacitation in sperm cells for short periods (Soriano-Úbeda 2019). Therefore, although bicarbonate may trigger some early capacitation-related changes, the medium used here is considered non-full capacitating in functional terms because it does not support the full sequence leading to capacitation or acrosome reaction.
Parrish JJ, Susko-Parrish J, Winer MA, First NL. Capacitation of bovine sperm by heparin. Biol Reprod. 1988 Jun;38(5):1171-80. doi: 10.1095/biolreprod38.5.1171. PMID: 3408784.
Soriano-Úbeda C, Romero-Aguirregomezcorta J, Matás C, Visconti PE, García-Vázquez FA. Manipulation of bicarbonate concentration in sperm capacitation media improves in vitro fertilisation output in porcine species. J Anim Sci Biotechnol. 2019 Mar 11;10:19. doi: 10.1186/s40104-019-0324-y. PMID: 30899459; PMCID: PMC6410524.
- Explicitly state that all experiments used straws from one bull/ejaculate.
We acknowledge the reviewer’s point regarding biological replication. All samples were obtained from frozen straws derived from a single bull and ejaculate; therefore, the data represent technical rather than biological replicates. This design was intentionally chosen to minimize inter-individual variability and to establish a controlled, mechanistic proof-of-concept of thapsigargin’s effects on Ca²⁺ homeostasis and sperm function. We have clarified this limitation in the revised manuscript and will include biological replication (≥3 bulls) in future studies to validate and generalize the findings. Please, see page 22, line 361.
- The temporal scheme (Fig. 1) is unclear: Tg was added for 2 min at alltime points, meaning t = 0 already includes Tg exposure. Provide an absolute timeline diagram.
Incubation with thapsigargin lasted a maximum of two minutes in each situation, during which motility, acrosome reaction, and actin polymerization were evaluated. There was no significant change in the parameters during this time period; therefore, the records analyzed correspond to the first few seconds of incubation with thapsigargin and are indicated in the same way as the respective controls.
- Missing vehicle control, viability control, and details of CASA video acquisition (fps, temperature, number of tracks analyzed).
We improved the paragraph by adding the information as follows: “Control and treated sperm samples (10 µL) were collected at each time point, placed on pre-warmed glass slides, and covered with a coverslip. The slides were maintained at 38 °C on a thermostated microscope stage and observed under transmitted light at 100× magnification using an LSM 800 Carl Zeiss microscope equipped with an Axio Observer Z1 camera. Videos were acquired with ZEN Blue software for 3–4 seconds at a frame rate of 63 frames/s. For each time point (0, 10, 20, and 30 min), 4–5 recordings were analyzed per experiment. Motility parameters were quantified using the OpenCASA plugin for ImageJ”. Please, see page 6, lines 118-121.
- FLUO-3 AM fluorescence was recorded in a plate reader without ratiometric calibration (no ionomycin/Fmax, no Mn²⁺ quench); therefore, results are semi-quantitative.
We agree with the reviewer that, because FLUO-3 AM is not a ratiometric fluorescence calcium indicator, the results obtained should be interpreted as semi-quantitative. Our aim, however, was not to determine absolute intracellular Ca2+ concentration but rather to compare relative changes in cytosolic Ca2+ levels between control and thapsigargin-treated samples.
This methodological approach is widely used in sperm physiology studies to detect relative variations in Ca2+ dynamics under different pharmacological conditions (Sánchez-Cárdenas et al, 2022; Durithala et al, 2022)
Sánchez-Cárdenas C, Romarowski A, Orta G, De la Vega-Beltrán JL, Martín-Hidalgo D, Hernández-Cruz A, Visconti PE, Darszon A. Starvation induces an increase in intracellular calcium and potentiates the progesterone-induced mouse sperm acrosome reaction. FASEB J. 2021 Apr;35(4):e21528. doi: 10.1096/fj.202100122R. PMID: 33742713; PMCID: PMC8441833.
Duritahala, Sakase M, Harayama H. Involvement of Ca2+-ATPase in suppressing the appearance of bovine helically motile spermatozoa with intense force prior to cryopreservation. J Reprod Dev. 2022 Jun 1;68(3):181-189. doi: 10.1262/jrd.2021-143. Epub 2022 Mar 3. PMID: 35236801; PMCID: PMC9184823.
We have now added a paragraph clarifying this point in the Materials and Methods (please, see page 8, lines 167-170) and discussion (please, see page 21, lines 337-339) sections.
Material and Methods
Assessment of Ca2+i
…”Variations in fluorescence intensity were used as an index of intracellular Ca²⁺ fluctuations. Because no ratiometric calibration was performed (no ionomycin/Fmax or Mn²⁺ quench/Fmin), data are expressed as relative fluorescence changes (ΔF/F₀) and represent semi-quantitative comparisons of cytosolic Ca²⁺ levels between experimental conditions”....
Discussion
Although the use of FLUO-3 AM without ratiometric calibration does not allow estimation of absolute intracellular Ca²⁺ concentrations, this approach is widely used to evaluate relative changes in Ca²⁺ dynamics in sperm cells. In our study, the increase in fluorescence observed after SERCA inhibition should therefore be interpreted as a relative rise in cytosolic Ca²⁺, consistent with previous findings in human and porcine sperm (Sánchez-Cárdenas et al, 2002; Duritahala et al, 2022).
- Immunofluorescence: co-staining with PSA/PNA and MitoTracker is essential to confirm localization.
Thank you for the observation. We added a merged image of SERCA staining and PSA. Please, see page 10, figure 1.
Results
- SERCA localization (Fig. 2): fluorescence is strongest in the equatorial/post-acrosomal region, not the acrosome; the claim of “reuptake into the acrosome” is inconsistent with the data.
We propose that SERCA is localised in the acrosomal region oriented toward the centre of the sperm head rather than the tip. To clarify this, we have included a superimposed image showing intact sperm acrosomes alongside SERCA localisation, highlighting its position in the acrosome oriented toward the equatorial region of the head. Please, see page 10, figure 1.
- Motility (Figs. 3–4): decreases in total/progressive motility with Tg likely reflect cytotoxicity, not physiological modulation. “Hyperactivation” inferred from increased ALH/VCL/BCF is speculative without 3D-tracking or energy expenditure data.
We agree with the reviewer. To highlight this point, we added a paragraph in the discussion section. …”One limitation in motility measures is that although CASA systems are often used in research and regular practices, CASA motility parameters are obtained by analyzing 2D trajectories of motion using videos of sperm movement. Recent studies proposed evaluating 3D sperm movements (helical movements with 3D rotation and 3D full-type hyperactivation) for a more precise analysis of sperm motility (Montoya et al, 2025)...”Please, see page 20, lines 316-320.
- Capacitation (Fig. 5) and Acrosome Reaction (Fig. 6): experimental timeline confounds interpretation; CTC and PSA concentrations inconsistent; replicate number too small (n = 3–4).
For all assays, Tg was applied for 2 min immediately before each sampling time point. We fixed the labeling error for PSA being the correct concentration 50 ug/ml. The n indicates technical replicates, but more than 200 cells for each condition were observed and classified into the categories, like non-capacitated, capacitated, and reacted (AR). Numbers of cells are indicated inside the bars.
- Actin polymerization (Fig. 7): the interpretation of “repolymerization at 30 min” lacks mechanistic support; no total actin quantification.
We appreciate this valuable comment. We agree that our interpretation of “repolymerization” at 30 min should be considered descriptive rather than mechanistic, since the fluorescence intensity of phalloidin staining reflects relative F-actin abundance but does not distinguish between polymerization and stabilization processes. Accordingly, the text has been revised to state that “a partial recovery of F-actin fluorescence was observed at 30 min” instead of “repolymerization.” We also acknowledge that total actin content was not quantified in the present study; therefore, our data represent relative changes in F-actin labeling intensity. We clarified this point in the section results. Please see page 17, lines 279-282.
- Ca²⁺ assay (Fig. 8): absence of calibration precludes quantitative comparison; fluorescence increase after Tg merely reflects blocked reuptake, not physiological signaling.
We thank the reviewer for this insightful comment. As we wrote above, the use of FLUO-3 AM without ratiometric calibration (no ionomycin/Fmax or Mn²⁺ quench/Fmin) limits quantitative interpretation of absolute intracellular Ca²⁺ concentrations. In the present study, fluorescence was used as a semi-quantitative indicator of relative changes in cytosolic Ca²⁺ between experimental conditions. We have clarified this explicitly in the Methods and Results sections. See page 8, lines 167-170.. We also agree that the increase in fluorescence after thapsigargin primarily reflects the inhibition of Ca²⁺ reuptake into intracellular stores rather than activation of a physiological signaling cascade. This interpretation is now stated in the Discussion, emphasizing that the assay reports functional SERCA inhibition and consequent changes in cytosolic Ca²⁺ handling, rather than dynamic second-messenger signaling. See page Page 21, line -334-337.
Future experiments using ratiometric probes (e.g., Fura-2 AM) or genetically encoded Ca²⁺ indicators will be required to quantify absolute Ca²⁺ levels and resolve compartment-specific kinetics.
Discussion
- Over-extends conclusions from phenomenological correlations.
Thank you. We rewrote the discussion based on the reviewer's suggestion and this observation.
- Contradiction: localization reported in the equatorial segment but discussed as acrosomal.
We agree with the reviewer. The expression is confusing, and we did not express the idea correctly. We suggest that SERCA is iin the acrosome region oriented toward the center of the head, not toward the tip. Unlike other stains, where localization is shown throughout the acrosome, here it is concentrated in an equatorial midline. To clarify this point, we have added a superimposed image showing intact sperm acrosomes and the location of SERCA in the acrosome, oriented toward the equatorial region of the head.
- The statement that SERCA “remains functional after cryopreservation” is speculative; fresh sperm were never tested.
We understand the observation. However, although fresh sperm were not compared here, SERCA is functional in the cryopreserved sperm studied. We mentioned this point in the discussion section. See page 2, lines 361 to 362.
- The possibility of SPCA involvement is acknowledged nowhere; this omission weakens mechanistic depth.
We thank the reviewer for this valuable observation. We agree that SPCA (secretory pathway Ca²⁺-ATPase) represents an additional Ca²⁺-transporting system that may contribute to calcium homeostasis in spermatozoa. While our experiments were designed to assess SERCA-related activity through thapsigargin inhibition, we acknowledge that SPCA isoforms, particularly SPCA1, are also expressed in mammalian sperm and may participate in acrosomal Ca²⁺ sequestration. We have added a statement in the Discussion noting that our results cannot exclude partial SPCA involvement and that further studies using specific inhibitors or molecular localization approaches are required to distinguish SERCA- and SPCA-mediated effects. Relevant references have been added to support this point. See page 21, lines 340- 344.
- Many paragraphs reiterate prior literature rather than critically integrate the present data.
We re-wrote the discussion to clarify these aspects.
- Rewrite with a cautious tone (“our findings suggest…”, “consistent with the hypothesis that…”).
We revised this point.
References
- Generally appropriate, but some typographical errors (“Journal of Androloy”) and inconsistent formatting
References have been checked
- Missing key references on SPCA, PMCA, and CatSper in bovine sperm.
References were added. Please, see page 21, line 342.
Comments on the Quality of English Language
The manuscript is generally understandable, but numerous grammatical and syntactic errors compromise fluency and precision. The text alternates between American and British English, with inconsistent capitalization of technical terms (e.g., Atpase vs. ATPase, acrosomal reaction vs. acrosome reaction). Sentence structure is sometimes verbose, and transitions between sections lack coherence.
A thorough professional English revision is required to correct terminology, improve readability, and ensure stylistic consistency throughout the text.
Thank you. Grammar has been checked.

Reviewer 2 Report
Comments and Suggestions for Authors
The article “Role of The Ca2+-Atpase Pump (Serca) in Capacitation and Acrosomal Reaction of Cryopreserved Bull Sperm Cells” by Rodríguez et al., is intended to prove the role of SERCA on functional bull sperm parameters; however, some issues should be pointed out:
1.- Line 29. Please change from “cytosolic calcium levels” to “intracellular calcium concentration [Ca2+]i" whenever is mentioned.
2.- Line 42. Please revise this sentence, specifically capacitation and acrosome reaction are independent process, since capacitation is ended therefore sperm can undergo the acrosome reaction.
3.- Line 85. Please add a proper reference to achieve capacitation and no-capacitation conditions
4.- Please, remove Figure 1
5.- A viability assay in needed to confirm the specific effect of thapsigargin on bull sperm
6.- Figure 2. To confirm the authors claims, about the presence and localization of sperm SERCA protein western-blot analysis is needed.
7.- Figure 8. A negative control is needed to validate the effect of thapsigargin on the [Ca2+]i e.g. NiCl2
8.- The presence of thapsigargin induce an increase of [Ca2+]i, however this effect is related to cellular stress, maybe the evaluation of ROS in presence of this compound will help to elucidate such effect, as well all changes in sperm motility parameters
Author Response
Reviewer 2
Comments and Suggestions for Authors
The article “Role of The Ca2+-Atpase Pump (Serca) in Capacitation and Acrosomal Reaction of Cryopreserved Bull Sperm Cells” by Rodríguez et al., is intended to prove the role of SERCA on functional bull sperm parameters; however, some issues should be pointed out:
1.- Line 29. Please change from “cytosolic calcium levels” to “intracellular calcium concentration [Ca2+]i" whenever is mentioned.
We have revised the expression in accordance with the reviewer’s recommendation. Kindly refer to page 2, lines 31-32, in the revised version of the manuscript.
2.- Line 42. Please revise this sentence, specifically capacitation and acrosome reaction are independent process, since capacitation is ended therefore sperm can undergo the acrosome reaction.
We thank the reviewer for this valuable observation. The sentence in the Introduction has been revised to clarify the events associated with capacitation and the acrosome reaction, with appropriate references added. Please see page 3, lines 45-46.
3.- Line 85. Please add a proper reference to achieve capacitation and no-capacitation conditions
Reference has been added. Please see page 4 line 84.
4.- Please, remove Figure 1
Figure 1 has been removed.
5.- A viability assay in needed to confirm the specific effect of thapsigargin on bull sperm
Thanks for the comment. We carried out the eosin-nigrosine assay to evaluate the viability of sperm cells under incubation with thapsigargin, DMO, and in the absence of drug. We added a paragraph in the material and methods, and results sections. Please, see pages 8-9, lines 173 to 187 in the methods section and pages 10-11, lines 210 to 213 in the results section.
6.- Figure 2. To confirm the authors claims, about the presence and localization of sperm SERCA protein western-blot analysis is needed.
We thank the reviewer for raising this important point regarding antibody specificity. The monoclonal antibody (Santa Cruz Biotechnology, sc-8095, clone F-1) was generated using a peptide derived from the N-terminus of human SERCA2, and it has been tested by the manufacturer (see manufacturer's datasheet: thttps://www.scbt.com/es/p/serca2-antibody-f-1).
We have confirmed through bioinformatic analysis that this specific immunogen region (residues 2-29) is 100% identical between the human (UniProt: ATP2A2_HUMAN (UniProt accession P16615) and bovine (UniProt accession A0AAA9T933) orthologs. This sequence conservation provides a strong theoretical basis for the expected cross-reactivity. Furthermore, the antibody has been validated by the manufacturer (see manufacturer's datasheet: https://www.scbt.com/es/p/serca2-antibody-f-1). Our experimental results provide direct evidence of its specificity. Importantly, our negative control (omitting the primary antibody) showed no signal, confirming the specificity of the observed fluorescence. Please see supplementary figure 1 and page 9, liness 199-201.
UniProt. (2025, October). ATP2A2_HUMAN (UniProt accession P16615) and ATP2A2_BOVIN (UniProt accession A0AAA9T933). UniProtKB protein knowledgebase. Retrieved October 16, 2025, from https://www.uniprot.org
7.- Figure 8. A negative control is needed to validate the effect of thapsigargin on the [Ca2+]i e.g. NiCl2
We appreciate the reviewer’s suggestion to use NiCl₂ as an additional control to block extracellular calcium entry. In our study, we aimed to evaluate the role of extracellular calcium in the thapsigargin-induced increase in cytosolic Ca²⁺. To this end, experiments were conducted in TALP medium depleted of Ca²⁺ and supplemented with 5 mM EGTA. This approach effectively chelates free extracellular calcium, preventing its influx into the cells. Under these conditions, thapsigargin treatment resulted in a decrease in fluorescence, indicating that the rise in cytosolic Ca²⁺ observed in calcium-containing medium is at least partially dependent on extracellular calcium. While NiCl₂ can block specific calcium entry channels, our experimental setup, using Ca²⁺-free medium and EGTA, provides a robust control for assessing the contribution of extracellular calcium and supports our conclusions regarding SERCA-mediated calcium mobilization.
8.- The presence of thapsigargin induce an increase of [Ca2+]i, however this effect is related to cellular stress, maybe the evaluation of ROS in presence of this compound will help to elucidate such effect, as well all changes in sperm motility parameters
We thank the reviewer for this insightful comment. We agree that thapsigargin can elicit cellular stress and potentially increase ROS generation, particularly in somatic cells under prolonged exposure (Yip et al., 2005; Ma et al., 2016; Wang et al., 2016). In our study, however, spermatozoa were exposed to thapsigargin for a short incubation period under controlled temperature and ionic conditions, minimising secondary oxidative effects. We have clarified this in the revised Discussion and added a statement acknowledging that future studies should assess ROS formation to further distinguish between specific Ca²⁺ dysregulation and stress-related responses. Please see pages 21-22, lines 352-360.
Wang C, Li T, Tang S, Zhao D, Zhang C, Zhang S, Deng S, Zhou Y, Xiao X. Thapsigargin induces apoptosis when autophagy is inhibited in HepG2 cells and both processes are regulated by ROS-dependent pathway. Environ Toxicol Pharmacol. 2016 Jan;41:167-79. doi: 10.1016/j.etap.2015.11.020. Epub 2015 Dec 9. PMID: 26708201.
Yip KH, Zheng MH, Steer JH, Giardina TM, Han R, Lo SZ, Bakker AJ, Cassady AI, Joyce DA, Xu J. Thapsigargin modulates osteoclastogenesis through the regulation of RANKL-induced signaling pathways and reactive oxygen species production. J Bone Miner Res. 2005 Aug;20(8):1462-71. doi: 10.1359/JBMR.050324. Epub 2005 Mar 28. PMID: 16007343.
Ma Z, Fan C, Yang Y, Di S, Hu W, Li T, Zhu Y, Han J, Xin Z, Wu G, Zhao J, Li X, Yan X. Thapsigargin sensitizes human esophageal cancer to TRAIL-induced apoptosis via AMPK activation. Sci Rep. 2016 Oct 12;6:35196. doi: 10.1038/srep35196. PMID: 27731378; PMCID: PMC5059685.

Reviewer 3 Report
Comments and Suggestions for Authors
In this study, the Authors investigated the role of SERCA in cryopreserved bovine spermatozoa, demonstrating that its activity significantly influences sperm motility, capacitation progression, actin polymerization, and intracellular calcium handling, ultimately affecting the timing of the acrosomal reaction.
The article is overall well-written, and the results are interesting.
However, I find the discussion somewhat diffuse, which makes it challenging for the reader to clearly identify the main findings and their comparison with existing literature.
Introducing a summarizing figure or schematic could greatly enhance clarity and highlight the key results and mechanistic insights.
While the discussion thoroughly addresses species-specific differences and the effects of SERCA inhibition, it could benefit from a more structured synthesis emphasizing the broader implications for cryopreserved bovine sperm function and fertilization efficiency.
In my opinion, such restructuring would help readers better appreciate how the observed cellular and molecular changes integrate into the overall physiological context.
Author Response
Reviewer 3
Comments and Suggestions for Authors
In this study, the Authors investigated the role of SERCA in cryopreserved bovine spermatozoa, demonstrating that its activity significantly influences sperm motility, capacitation progression, actin polymerization, and intracellular calcium handling, ultimately affecting the timing of the acrosomal reaction.
The article is overall well-written, and the results are interesting.
However, I find the discussion somewhat diffuse, which makes it challenging for the reader to clearly identify the main findings and their comparison with existing literature.
Introducing a summarizing figure or schematic could greatly enhance clarity and highlight the key results and mechanistic insights.
While the discussion thoroughly addresses species-specific differences and the effects of SERCA inhibition, it could benefit from a more structured synthesis emphasizing the broader implications for cryopreserved bovine sperm function and fertilization efficiency.
In my opinion, such restructuring would help readers better appreciate how the observed cellular and molecular changes integrate into the overall physiological context.
We thank the reviewer for the suggestion. We have revised and strengthened the Discussion to incorporate the relevant findings. Please see the Discussion section, pages 19–22.

Reviewer 4 Report
Comments and Suggestions for Authors
This manuscript investigates the role of the Sarco/Endoplasmic Reticulum Ca2+-ATPase (SERCA) pump in several key physiological processes of cryopreserved bull spermatozoa. This work also uses thapsigargin (Tg), a specific SERCA inhibitor. The study describes that SERCA is functional in cryopreserved sperm. When is inhibited it leads to reduced motility, the induction of hyperactivated-like movement patterns, and an acceleration of the acrosome reaction. The study addresses an important topic, as understanding the subcellular mechanisms governing sperm function after cryopreservation is crucial for improving artificial insemination techniques. Nevertheless, there are several major weaknesses in its experimental design that hamper data interpretation and highlight some drawbacks. The most significant of these is the use of sperm from a single bull, which severely limits the generalizability of the findings. I have some specific comments:
- The most critical problem in this study is that all experiments were conducted using cryopreserved sperm originating from a single bull. The authors used multiple straws and many cells were analyzed but this constitutes a technical replication and not biological replications. It is known that sperm physiology and cryotolerance can vary between individuals. The experiments should include samples from multiple bulls. This limitation prevents the findings from being generalized. The conclusions should be tempered accordingly, framing the results as preliminary.
- The authors report that SERCA inhibition with Tg reduces total and progressive motility across all time points. Simultaneously, they observe increases in parameters like VCL, BCF, and ALH at later time points, which they interpret as the induction of a hyperactivation pattern. I am not sure of this interpretation since hyperactivation is a physiological state characterized by vigorous motility that is critical for fertilization. However, the concurrent and significant decrease in the overall percentage of motile sperm may not be due to functional hyperactivation but rather a pathological response to cytotoxic levels of intracellular calcium. The erratic movements could be indicative of cell distress and membrane dysfunction leading to cell death. The authors should know that calcium dysregulation may cause a loss of coordinated flagellar movement and viability.
- The lack of a parallel set of experiments on fresh ejaculated sperm from the same bull(s) does not allow the study of the specific impact of the cryopreservation. The authors acknowledge that cryopreservation itself can induce capacitation-like changes ("cryo-capacitation"). Therefore, it is plausible that SERCA's role or its susceptibility to inhibition is different in cryopreserved cells compared to fresh ones. Overall, the current design only demonstrates that SERCA is functional post-thaw but not how its role might be altered by the freezing and thawing process.
- The study uses a single 10 µM concentration of thapsigargin. There is no justification for this choice. A dose-response experiment would be necessary to confirm that the effects are not due to off-target or toxic effects at a high concentration. Atually, I am not convinced that the observed decrease in total motility is not a potential toxic effect.
- The authors compare Tg-treated samples to a "control" in capacitating medium. However, a vehicle control (i.e., the solvent used to dissolve Tg, presumably DMSO) is not mentioned.
- For the SERCA localization experiments, the authors show immunofluorescence images but provide no validation of the antibody's specificity in bull sperm (e.g., via Western blot). This is particularly important given that their localization to the equatorial segment contradicts another study in cryopreserved bull sperm that found SERCA in the midpiece. Please provide more validation.
- The data shows that after an early depolymerization event in the Tg group at 10 minutes, F-actin fluorescence appears to recover at 20 and 30 minutes. This "re-polymerization" after the acrosome reaction has supposedly been triggered is counterintuitive and is not explained or discussed. It raises questions about the sequence of events and the final state of the sperm.
Author Response
This manuscript investigates the role of the Sarco/Endoplasmic Reticulum Ca2+-ATPase (SERCA) pump in several key physiological processes of cryopreserved bull spermatozoa. This work also uses thapsigargin (Tg), a specific SERCA inhibitor. The study describes that SERCA is functional in cryopreserved sperm. When is inhibited it leads to reduced motility, the induction of hyperactivated-like movement patterns, and an acceleration of the acrosome reaction. The study addresses an important topic, as understanding the subcellular mechanisms governing sperm function after cryopreservation is crucial for improving artificial insemination techniques. Nevertheless, there are several major weaknesses in its experimental design that hamper data interpretation and highlight some drawbacks. The most significant of these is the use of sperm from a single bull, which severely limits the generalizability of the findings. I have some specific comments:
- The most critical problem in this study is that all experiments were conducted using cryopreserved sperm originating from a single bull. The authors used multiple straws and many cells were analyzed but this constitutes a technical replication and not biological replications. It is known that sperm physiology and cryotolerance can vary between individuals. The experiments should include samples from multiple bulls. This limitation prevents the findings from being generalized. The conclusions should be tempered accordingly, framing the results as preliminary.
We acknowledge the reviewer’s point regarding biological replication. All samples were obtained from frozen straws derived from a single bull and ejaculate; therefore, the data represent technical rather than biological replicates. This design was intentionally chosen to minimise inter-individual variability and to establish a controlled, mechanistic proof-of-concept of thapsigargin’s effects on Ca²⁺ homeostasis and sperm function. We have clarified this limitation in the revised manuscript and noted that future studies should include biological replicates to validate and generalise the findings. Please see the Discussion section, page 22, lines 361–363.
- The authors report that SERCA inhibition with Tg reduces total and progressive motility across all time points. Simultaneously, they observe increases in parameters like VCL, BCF, and ALH at later time points, which they interpret as the induction of a hyperactivation pattern. I am not sure of this interpretation since hyperactivation is a physiological state characterized by vigorous motility that is critical for fertilization. However, the concurrent and significant decrease in the overall percentage of motile sperm may not be due to functional hyperactivation but rather a pathological response to cytotoxic levels of intracellular calcium. The erratic movements could be indicative of cell distress and membrane dysfunction leading to cell death. The authors should know that calcium dysregulation may cause a loss of coordinated flagellar movement and viability.
Thanks to the reviewer for the suggestion. We carried out the eosin-nigrosin assay to evaluate the viability of sperm cells under incubation with thapsigargin, DSMO, and in the absence of drug. No significant differences in the proportion of viable cells were detected among the groups (p > 0.05). We added a paragraph in the material and methods sections, results, and a supplementary figure. Please, see pages 8-9, lines 173–186 in the methods section and pages 10-11, lines 210–213 in the results section.
- The lack of a parallel set of experiments on fresh ejaculated sperm from the same bull(s) does not allow the study of the specific impact of the cryopreservation. The authors acknowledge that cryopreservation itself can induce capacitation-like changes ("cryo-capacitation"). Therefore, it is plausible that SERCA's role or its susceptibility to inhibition is different in cryopreserved cells compared to fresh ones. Overall, the current design only demonstrates that SERCA is functional post-thaw but not how its role might be altered by the freezing and thawing process.
Thank you for this observation. We re-wrote the discussion clarifying this aspect.
- The study uses a single 10 µM concentration of thapsigargin. There is no justification for this choice. A dose-response experiment would be necessary to confirm that the effects are not due to off-target or toxic effects at a high concentration. Atually, I am not convinced that the observed decrease in total motility is not a potential toxic effect.
The dose selection was based on the analysis by Ho and Suarez on bovine sperm (Ho and Suarez, 2001), Williams and Ford (Williams and Ford, 2003), and Durithala et al. (2022). In Ho and Suarez (2001), the authors performed a dose–response curve. Based on their results, we selected 10 μM, a moderate dose that is sufficiently effective to evaluate changes in motility and the acrosomal reaction, while allowing a reasonably short-term incubation to minimise potential off-target effects.
We have revised the manuscript to clarify that the effects observed at this concentration may include off-target or cytotoxic components, and we now present our conclusions as mechanistic rather than exclusively SERCA-specific. Future work could include a full dose–response analysis and alternative SERCA inhibitors (CPA, BHQ) to strengthen the causal interpretation. Please see the Discussion section, page 22, lines 363–365.
References
Ho HC, Suarez SS. An inositol 1,4,5-trisphosphate receptor-gated intracellular Ca(2+) store is involved in regulating sperm hyperactivated motility. Biol Reprod. 2001 Nov;65(5):1606-15. doi: 10.1095/biolreprod65.5.1606. PMID: 11673282.
Williams KM, Ford WC. Effects of Ca-ATPase inhibitors on the intracellular calcium activity and motility of human spermatozoa. Int J Androl. 2003 Dec;26(6):366-75. doi: 10.1111/j.1365-2605.2003.00438.x. PMID: 14636222.
Duritahala, Sakase M, Harayama H. Involvement of Ca2+-ATPase in suppressing the appearance of bovine helically motile spermatozoa with intense force prior to cryopreservation. J Reprod Dev. 2022 Jun 1;68(3):181-189. doi: 10.1262/jrd.2021-143. Epub 2022 Mar 3. PMID: 35236801; PMCID: PMC9184823.
- The authors compare Tg-treated samples to a "control" in capacitating medium. However, a vehicle control (i.e., the solvent used to dissolve Tg, presumably DMSO) is not mentioned.
Thank you. We clarify this aspect. Please, see page 8 line 175 in methods and pages 10-11 lines 211-213 in results section.
- For the SERCA localization experiments, the authors show immunofluorescence images but provide no validation of the antibody's specificity in bull sperm (e.g., via Western blot). This is particularly important given that their localization to the equatorial segment contradicts another study in cryopreserved bull sperm that found SERCA in the midpiece. Please provide more validation.
The monoclonal antibody (Santa Cruz Biotechnology, sc-8095, clone F-1) was generated using a peptide derived from the N-terminus of human SERCA2, and it has been tested by the manufacturer (see manufacturer's datasheet: thttps://www.scbt.com/es/p/serca2-antibody-f-1).
We have confirmed through bioinformatic analysis that this specific immunogen region (residues 2-29) is 100% identical between the human (UniProt: ATP2A2_HUMAN (UniProt accession P16615) and bovine (UniProt accession A0AAA9T933) orthologs. This sequence conservation provides a strong theoretical basis for the expected cross-reactivity. Furthermore, the antibody has been validated by the manufacturer (see manufacturer's datasheet: https://www.scbt.com/es/p/serca2-antibody-f-1). Our experimental results provide direct evidence of its specificity. Importantly, our negative control (omitting the primary antibody) showed no signal, confirming the specificity of the observed fluorescence. See supplementary figure 1. Moreover, we included a merged image combining the anti-SERCA antibody and PSA staining to visualise both SERCA and the acrosome in spermatozoa. This control has been added to Figure 1 in the Results section (page 10), to enhance clarity.
UniProt. (2025, October). ATP2A2_HUMAN (UniProt accession P16615) and ATP2A2_BOVIN (UniProt accession A0AAA9T933). UniProtKB protein knowledgebase. Retrieved October 16, 2025, from https://www.uniprot.org
- The data shows that after an early depolymerization event in the Tg group at 10 minutes, F-actin fluorescence appears to recover at 20 and 30 minutes. This "re-polymerization" after the acrosome reaction has supposedly been triggered is counterintuitive and is not explained or discussed. It raises questions about the sequence of events and the final state of the sperm.
We thank the reviewer for this insightful comment. We agree that the apparent recovery of F-actin fluorescence at 20–30 min should not be interpreted as definitive “re-polymerization” after the acrosome reaction. Instead, it likely reflects a partial restoration or reorganization of actin structures in spermatozoa that did not undergo complete acrosomal exocytosis, or stabilization of residual cortical actin in non-reacted cells. We have revised the Results and Discussion to clarify that these observations represent relative changes in F-actin labeling intensity rather than mechanistically confirmed polymerization. We now acknowledge that this finding may indicate heterogeneity in sperm responses over time, and that further work combining F/G-actin quantification or live-cell imaging will be required to elucidate the sequence of actin remodeling events.
Round 2
Reviewer 1 Report
Comments and Suggestions for Authors
Thank you for the revised version of the manuscript and the detailed point-by-point response. The study addresses an interesting question regarding Ca²⁺ handling in cryopreserved bovine spermatozoa, and the overall quality of the text, structure, and statistical treatment has clearly improved compared to the previous version. Many of the issues raised in the first round were addressed, including: (i) clarification of the experimental hypothesis; (ii) inclusion of a viability/vehicle control; (iii) correction of fluorophore and PSA concentration inconsistencies; (iv) improved statistical analysis using mixed models with FDR correction; (v) explicit acknowledgment that all experiments were performed using straws from a single ejaculate of one bull; and (vi) substantial rewriting of the Discussion in a more cautious tone.
However, there are still several points that need further revision before the manuscript can be considered ready. Most of these are clarifications or consistency adjustments, and they do not require new experiments. They are listed below.
- Specificity of thapsigargin and strength of mechanistic claims
In the current version, some statements still suggest a direct, specific, and causal role for SERCA in sperm capacitation and acrosome reaction. This is not fully supported by the data, because:
- all functional data rely on a single inhibitor (thapsigargin) at 10 µM;
- this concentration is orders of magnitude higher than the nanomolar range typically used for selective SERCA inhibition in other systems, and such high doses are known to increase the likelihood of off-target and potentially cytotoxic effects.
You have partially addressed this in the Discussion, where the language is now more cautious, but there are still two places where the wording remains too strong:
(a) In the Abstract, you currently conclude that “SERCA is functional in cryopreserved spermatozoa and accelerates events leading to the acrosome reaction.”
(b) In the Results, one subsection/header still refers to SERCA as “essential” for Ca²⁺ regulation.
Please revise these to use hypothesis-based language. For example:
- “Our findings suggest that a thapsigargin-sensitive Ca²⁺ pump consistent with SERCA activity remains responsive in cryopreserved spermatozoa and may contribute to Ca²⁺ handling, motility changes, and premature acrosomal exocytosis.”
- Replace “SERCA is essential for…” with wording such as “Thapsigargin-sensitive Ca²⁺ sequestration modulates intracellular Ca²⁺ levels…”
In addition, please include an explicit sentence in the Discussion noting that the 10 µM thapsigargin concentration used here is substantially higher than commonly reported nanomolar doses for selective SERCA inhibition, and therefore off-target and cytotoxic components cannot be excluded. This clarification is important for transparency.
- Biological replication and scope of the conclusions
You now clearly state in both Methods and Discussion that all samples came from a single ejaculate of one bull, and you frame the work as mechanistic/proof-of-concept. This is an important improvement.
However, the Conclusions (and parts of the Abstract) still generalize to “cryopreserved bovine spermatozoa” in broad terms. Please adjust the final statements of the Abstract and Conclusions to reflect that these findings were obtained in ejaculates from a single bull, under the specific cryopreservation and incubation conditions tested here. This avoids over-generalization and aligns with the limitation you already acknowledge in the Discussion.
- Vehicle control and viability
You added an eosin–nigrosin viability assay comparing untreated, DMSO-treated, and thapsigargin-treated sperm, and you report no significant differences in viability among these groups. You also refer to this in the Methods/Results and provide a supplementary figure. This is an excellent addition.
To close the loop methodologically, please add one clarifying sentence in either the Results or Materials and Methods explicitly stating that, because the vehicle (DMSO) did not affect viability relative to untreated controls, subsequent functional comparisons (motility, capacitation, acrosome reaction) were performed using untreated sperm as the control condition rather than a separate vehicle control. This logical link is already implied in your response to reviewers but should be stated in the manuscript.
- SERCA localization and antibody validation
You indicate in your response that you added (i) a merged image of SERCA staining with an acrosomal marker (PSA) to show the spatial relationship between SERCA signal and the acrosome, and (ii) a negative control omitting the primary antibody. The negative control is indeed described, and this is appreciated.
However, in the current version of the manuscript/PDF, the main figure still shows only the SERCA channel and does not include the merged SERCA+PSA image that you mention in your response. Please ensure that:
- the merged SERCA/PSA (or equivalent acrosomal marker) image is actually included in the figure panel or in the Supplementary Figures, and
- the figure legend explicitly states that SERCA immunofluorescence is shown alongside the acrosomal marker to illustrate relative localization.
Additionally, in your response to reviewers you describe a sequence alignment showing that the immunogen region of the anti-SERCA2 antibody (Santa Cruz, clone F-1) is 100% identical between the human and bovine orthologues, which supports expected cross-reactivity. This useful justification does not currently appear in the manuscript. Please insert one sentence in either Materials and Methods (Immunofluorescence) or Discussion summarizing this sequence identity and rationale for antibody specificity.
Finally, please add one sentence in the Discussion acknowledging that, although the observed signal is consistent with SERCA localization in the acrosomal/equatorial region, full organelle-level assignment is still limited by the lack of simultaneous mitochondrial markers (e.g., MitoTracker) and by the absence of isotype- or peptide-blocked controls. This keeps the interpretation appropriately cautious.
- Composition and terminology of “non-capacitating” vs. “capacitating” media
In your response you provide a conceptual justification for calling one medium “non-capacitating,” noting that bicarbonate alone can trigger early capacitation-like events but is not sufficient to drive full capacitation and acrosome reaction under short incubations. This clarification is scientifically important because, as originally noted, bicarbonate and BSA are generally considered capacitating factors.
In the current Materials and Methods, the exact compositions of the “non-capacitating” TALP and the “capacitating” TALP are not clearly distinguished in a way that lets the reader understand which components differ (bicarbonate, BSA, Ca²⁺, etc.), and why one is functionally considered “non-capacitating” in this study.
Please revise the Methods to:
- explicitly list the full composition of each medium;
- state clearly which medium you define as “non-capacitating” and which as “capacitating”;
- add one explanatory sentence indicating that the “non-capacitating” formulation, although it may contain bicarbonate, does not support the full cascade of capacitation/acrosome reaction within the short incubation window used here, and is therefore considered functionally non-capacitating in this experimental context.
This will resolve the apparent internal inconsistency.
- Experimental timeline / timing of thapsigargin exposure
One of the initial concerns was that the temporal design was difficult to follow (e.g., thapsigargin was added for 2 min before each time point, so “time 0” already includes Tg exposure, etc.). The revised text is more explicit, but it is still relatively hard for a reader to reconstruct the order of additions (caffeine, progesterone, thapsigargin), sampling points (0, 10, 20, 30 min), and which aliquots were used for which assays.
To improve clarity and reproducibility, please include a simple schematic timeline (as a new figure or supplementary figure) illustrating:
- thawing / initial incubation,
- addition of caffeine,
- addition of progesterone,
- sampling at 0, 10, 20, and 30 min,
- 2 min exposure ± thapsigargin immediately prior to motility / capacitation / acrosome / actin / Ca²⁺ readout.
This visual summary will significantly help readers interpret the functional data sets (motility, CTC patterns, PSA labeling, actin staining, Ca²⁺ signal).
- Statistical analysis and presentation of significance
It is positive that you now report using linear mixed models (MixedLM, statsmodels) with FDR correction, and that you provide Supplementary Table 1 and Supplementary Figure 2 (heatmap) as global statistical outputs. This addresses the previous concern about inflated type I error.
However, in the main figures you still annotate significance using pairwise tests (t-tests/ANOVA) without clearly stating how these p-values relate to the FDR-controlled analysis.
To avoid any perception of selective reporting, please add to the “Statistical analysis” section a brief clarifying sentence along the lines of:
“Overall effects of treatment and time were evaluated using a linear mixed model with false discovery rate (FDR) correction (see Supplementary Table 1 and Supplementary Figure 2). Pairwise significance markers indicated in the main figures refer to the specific post hoc comparisons described in each legend.”
This makes the statistical approach transparent.
- Minor language/typographical points
The English has improved substantially. Please make the following minor corrections:
- In the Results text describing actin dynamics, there is a sentence referring to “ERCA inhibition”. This appears to be a typographical error and should read “SERCA inhibition”.
- Please check for consistency in spelling (American vs. British English). For instance, “polymerization / polymerisation”, “organization / organisation”, etc. The manuscript currently mixes both styles. The journal typically prefers internal consistency within a manuscript.
- Ensure consistent formatting of Ca²⁺-ATPase, SERCA, and related abbreviations throughout, and verify that all abbreviations used in the figures (e.g., VSL, VCL, WOB, ALH, BCF, FD) appear in the list of abbreviations / figure legends.
The English language in the revised manuscript is generally understandable, but it still requires careful polishing to meet publication standards. The overall structure and clarity have improved compared with the first version, yet several grammatical and stylistic inconsistencies remain. The text alternates between British and American English, showing variations such as “polymerisation” versus “polymerization” and “behaviour” versus “behavior,” which should be standardized throughout. Technical terminology is sometimes inconsistently formatted or capitalized — for example, Ca²⁺-ATPase, acrosome reaction, and abbreviations like VSL, VCL, and WOB are not uniformly presented in the text and figure legends.
Certain sentences are overly long or contain redundant phrases, which affects fluency and precision. Occasional grammatical issues, such as missing articles, subject–verb disagreements, or misplaced modifiers, also reduce readability. The tone has improved and is now more cautious, but some expressions still sound too assertive given the limitations of the data — for instance, statements implying direct causality should be rephrased to conditional forms (“may contribute to,” “suggests that”). Minor typographical errors remain, including the term “ERCA inhibition” instead of SERCA inhibition, and some units and symbols could be standardized according to journal style.
In summary, the manuscript is clearly written enough to follow the scientific reasoning, but it would benefit from a final round of professional English editing to ensure grammatical accuracy, stylistic consistency, and a smooth, professional tone across all sections.
Author Response
Thank you for the revised version of the manuscript and thedetailed point-by-point response. The study addresses aninteresting question regarding Ca²⁺ handling in cryopreservedbovine spermatozoa, and the overall quality of the text, structure,and statistical treatment has clearly improved compared to theprevious version. Many of the issues raised in the first roundwere addressed, including: (i) clarification of the experimentalhypothesis; (ii) inclusion of a viability/vehicle control; (iii) improved statistical analysis using mixed models with FDRcorrection; (v) explicit acknowledgment that all experiments wereperformed using straws from a single ejaculate of one bull; and(vi) substantial rewriting of the Discussion in a more cautioustone.
However, there are still several points that need further revision before the manuscript can be considered ready. Most of these are clarifications or consistency adjustments, and they do not require new experiments. They are listed below.
- Specificity of thapsigargin and strength of mechanistic claims
In the current version, some statements still suggest a direct, specific, and causal role for SERCA in sperm capacitation and acrosome reaction. This is not fully supported by the data,
because: all functional data rely on a single inhibitor (thapsigargin) at 10
µM; this concentration is orders of magnitude higher than the nanomolar range typically used for selective SERCA inhibition in other systems, and such high doses are known to increase the likelihood of off-target and potentially cytotoxic effects.
You have partially addressed this in the Discussion, where the language is now more cautious, but there are still two places where the wording remains too strong:
(a) In the Abstract, you currently conclude that “SERCA is functional in cryopreserved spermatozoa and accelerates events leading to the acrosome reaction.”
(b) In the Results, one subsection/header still refers to SERCA as “essential” for Ca²⁺ regulation.
Please revise these to use hypothesis-based language. For example: “Our findings suggest that a thapsigargin-sensitive Ca²⁺ pump consistent with SERCA activity remains responsive in cryopreserved spermatozoa and may contribute to Ca²⁺ handling, motility changes, and premature acrosomal exocytosis.” Replace “SERCA is essential for…” with wording such as
“Thapsigargin-sensitive Ca²⁺ sequestration modulates intracellular Ca²⁺ levels…”
In addition, please include an explicit sentence in the Discussion noting that the 10 µM thapsigargin concentration used here is substantially higher than commonly reported nanomolar doses for selective SERCA inhibition, and therefore off-target and cytotoxic components cannot be excluded. This clarification is important for transparency.
We thank the reviewer for this valuable observation. The statements in the Abstract and Results have been revised in accordance with the reviewer’s suggestions to adopt hypothesis-based wording. Specifically, the concluding sentence of the Abstract (page 1, lines 29–33) and the subsection header in the Results (page 12, lines 375–376) now use phrasing consistent with “Thapsigargin-sensitive Ca²⁺ sequestration modulates intracellular Ca²⁺ levels in cryopreserved bull spermatozoa” In addition, a clarifying sentence has been added to the Discussion (page 14, lines 475–478) acknowledging that the 10 µM thapsigargin concentration used here is higher than the nanomolar range typically applied for selective SERCA inhibition and that potential off-target or cytotoxic effects cannot be excluded.
- Biological replication and scope of the conclusions
You now clearly state in both Methods and Discussion that all samples came from a single ejaculate of one bull, and you frame the work as mechanistic/proof-of-concept. This is an important improvement.
However, the Conclusions (and parts of the Abstract) still generalize to “cryopreserved bovine spermatozoa” in broad terms. Please adjust the final statements of the Abstract and
Conclusions to reflect that these findings were obtained in ejaculates from a single bull, under the specific cryopreservation and incubation conditions tested here. This avoids overgeneralization and aligns with the limitation you already acknowledge in the Discussion.
We have modified the sentences in the abstract and discussion sections to avoid generalizations. Please refer to page 1, line 30, in the abstract, and page 14, lines 481-482 and lines 489-490, at the end of the discussion, for clarification of these aspects.
- Vehicle control and viability
You added an eosin–nigrosin viability assay comparing untreated, DMSO-treated, and thapsigargin-treated sperm, and you report no significant differences in viability among these
groups. You also refer to this in the Methods/Results and provide
a supplementary figure. This is an excellent addition. To close the loop methodologically, please add one clarifying sentence in either the Results or Materials and Methods explicitly
stating that, because the vehicle (DMSO) did not affect viability relative to untreated controls, subsequent functional comparisons (motility, capacitation, acrosome reaction) were performed using untreated sperm as the control condition rather than a separate
vehicle control. This logical link is already implied in your response to reviewers but should be stated in the manuscript.
Thanks for the suggestion. A clarifying sentence has been added to the Results section to explicitly state that, as DMSO treatment did not affect sperm viability relative to untreated controls, subsequent functional analyses (motility, capacitation, and acrosome reaction) were conducted using untreated sperm as the control condition. Please see page 6, lines 246–251.
- SERCA localization and antibody validation
You indicate in your response that you added (i) a merged image of SERCA staining with an acrosomal marker (PSA) to show the spatial relationship between SERCA signal and the acrosome, and (ii) a negative control omitting the primary antibody. The
negative control is indeed described, and this is appreciated.
However, in the current version of the manuscript/PDF, the main figure still shows only the SERCA channel and does not include the merged SERCA+PSA image that you mention in your response. Please ensure that: the merged SERCA/PSA (or equivalent acrosomal marker) image is actually included in the figure panel or in the
Supplementary Figures, and the figure legend explicitly states that SERCA
immunofluorescence is shown alongside the acrosomal marker to illustrate relative localization.
We appreciate the reviewer’s careful observation. The merged image showing SERCA staining alongside the acrosomal marker (PSA) is shown in Figure 1; however, the original legend may have been unclear. The legend has now been revised to explicitly indicate that the figure presents SERCA immunofluorescence alongside PSA staining to illustrate their relative localization (see page 6, figure 1). In addition, the supplementary figure includes the control images showing cells without the primary antibody, SERCA staining, the corresponding transmitted-light image, and the merged image. The legend for this supplementary figure has also been updated for clarity (see page 15, lines 506-512, supplementary figure 2).
Additionally, in your response to reviewers you describe a sequence alignment showing that the immunogen region of the anti-SERCA2 antibody (Santa Cruz, clone F-1) is 100% identical between the human and bovine orthologues, which supports expected cross-reactivity. This useful justification does not currently appear in the manuscript. Please insert one sentence in either Materials and Methods (Immunofluorescence) or
Discussion summarizing this sequence identity and rationale for
antibody specificity.
Finally, please add one sentence in the Discussion acknowledging that, although the observed signal is consistent with SERCA localization in the acrosomal/equatorial region, full
organelle-level assignment is still limited by the lack of simultaneous mitochondrial markers (e.g., MitoTracker) and by the absence of isotype- or peptide-blocked controls. This keeps the interpretation appropriately cautious.
Thanks for the recommendation. A paragraph summarising the sequence alignment and supporting antibody specificity has been added to the Materials and Methods section (see page 3, lines 105–109). In addition, a sentence has been incorporated into the Discussion acknowledging that, although the observed fluorescence pattern is consistent with SERCA localisation in the acrosomal and equatorial regions, definitive organelle-level assignment remains limited by the absence of simultaneous mitochondrial markers and isotype- or peptide-blocked controls. Please, see page 13, lines 412-416.
- Composition and terminology of “non-capacitating” vs. ”capacitating” media
In your response you provide a conceptual justification for calling one medium “non-capacitating,” noting that bicarbonate alone can trigger early capacitation-like events but is not sufficient to drive full capacitation and acrosome reaction under short
incubations. This clarification is scientifically important because, as originally noted, bicarbonate and BSA are generally considered capacitating factors.
In the current Materials and Methods, the exact compositions of the “non-capacitating” TALP and the “capacitating” TALP are not clearly distinguished in a way that lets the reader understand which components differ (bicarbonate, BSA, Ca²⁺, etc.), and why
one is functionally considered “non-capacitating” in this study.
Please revise the Methods to:
explicitly list the full composition of each medium; state clearly which medium you define as “non-capacitating” and which as “capacitating”;
add one explanatory sentence indicating that the “noncapacitating” formulation, although it may contain bicarbonate, does not support the full cascade of capacitation/acrosome reaction within the short incubation window used here, and is therefore considered functionally non-capacitating in this experimental context.
This will resolve the apparent internal inconsistency.
We appreciate the reviewer’s constructive suggestions. The full composition of both media has now been explicitly detailed, and each medium is clearly identified as “non-capacitating” or “capacitating.” In addition, an explanatory sentence has been added to clarify that, although the non-capacitating TALP contains bicarbonate, it does not support the complete sequence of capacitation or acrosome reaction within the short incubation period used here and is therefore considered functionally non-capacitating in this experimental context. Please, see pages 2 and 3, lines 86-97.
- Experimental timeline / timing of thapsigargin exposure
One of the initial concerns was that the temporal design was difficult to follow (e.g., thapsigargin was added for 2 min before each time point, so “time 0” already includes Tg exposure, etc.).
The revised text is more explicit, but it is still relatively hard for a reader to reconstruct the order of additions (caffeine, progesterone, thapsigargin), sampling points (0, 10, 20, 30 min), and which aliquots were used for which assays.
To improve clarity and reproducibility, please include a simple schematic timeline (as a new figure or supplementary figure)
illustrating:
thawing / initial incubation,
addition of caffeine,
addition of progesterone,
sampling at 0, 10, 20, and 30 min,
2 min exposure ± thapsigargin immediately prior to motility /capacitation / acrosome / actin / Ca²⁺ readout.
This visual summary will significantly help readers interpret thefunctional data sets (motility, CTC patterns, PSA labeling, actin staining, Ca²⁺ signal).
We thank the reviewer for this helpful suggestion. A schematic timeline illustrating the sequence of treatments (thawing, caffeine and progesterone additions, thapsigargin exposures) and sampling points has been added to clarify the experimental design. This figure is Supplementary Figure 1 and is referenced in the Materials and Methods section for clarity. Please, see page 3, lines 132-133.
- Statistical analysis and presentation of significance
It is positive that you now report using linear mixed models (MixedLM, statsmodels) with FDR correction, and that you provide Supplementary Table 1 and Supplementary Figure 2 (heatmap) as global statistical outputs. This addresses the previous concern about inflated type I error.
However, in the main figures you still annotate significance using pairwise tests (t-tests/ANOVA) without clearly stating how these p-values relate to the FDR-controlled analysis.
To avoid any perception of selective reporting, please add to the “Statistical analysis” section a brief clarifying sentence along the lines of:
“Overall effects of treatment and time were evaluated using a linear mixed model with false discovery rate (FDR) correction (see Supplementary Table 1 and Supplementary Figure 2).
Pairwise significance markers indicated in the main figures refer to the specific post hoc comparisons described in each legend.” This makes the statistical approach transparent.
We added a sentence to clarify this point. Please, see page 5, lines 214-217.
- Minor language/typographical points
The English has improved substantially. Please make the following minor corrections:
In the Results text describing actin dynamics, there is a sentence referring to “ERCA inhibition”. This appears to be a typographical error and should read “SERCA inhibition”.
Please check for consistency in spelling (American vs. British English). For instance, “polymerization / polymerisation”, “organization / organisation”, etc. The manuscript currently mixes both styles. The journal typically prefers internal consistency
within a manuscript.
Ensure consistent formatting of Ca²⁺-ATPase, SERCA, and related abbreviations throughout, and verify that all abbreviations used in the figures (e.g., VSL, VCL, WOB, ALH, BCF, FD) appear in the list of abbreviations / figure legends.
The English language in the revised manuscript is generally understandable, but it still requires careful polishing to meet publication standards. The overall structure and clarity have improved compared with the first version, yet several grammatical and stylistic inconsistencies remain. The text alternates between British and American English, showing variations such as “polymerisation” versus “polymerization” and “behaviour” versus “behavior,” which should be standardized
throughout. Technical terminology is sometimes inconsistently formatted or capitalized — for example, Ca²⁺-ATPase, acrosome reaction, and abbreviations like VSL, VCL, and WOB are not uniformly presented in the text and figure legends.
Certain sentences are overly long or contain redundant phrases, which affects fluency and precision. Occasional grammatical issues, such as missing articles, subject–verb disagreements, or misplaced modifiers, also reduce readability. The tone has
improved and is now more cautious, but some expressions still sound too assertive given the limitations of the data — for instance, statements implying direct causality should be rephrased to conditional forms (“may contribute to,” “suggests that”). Minor typographical errors remain, including the term “ERCA inhibition” instead of SERCA inhibition, and some units and symbols could be standardized according to journal style. In summary, the manuscript is clearly written enough to follow the scientific reasoning, but it would benefit from a final round of professional English editing to ensure grammatical accuracy, stylistic consistency, and a smooth, professional tone across all sections
Improvements have been made in accordance with the suggestions.
Reviewer 2 Report
Comments and Suggestions for Authors
No further comments
Author Response
No further comments
Thanks to the reviewer for their valuable review of the manuscript.
Reviewer 3 Report
Comments and Suggestions for Authors
The authors have satisfactorily addressed all my previous concerns regarding the Discussion section. However, I still believe that including a summarizing figure would help highlight the key results and mechanistic insights. It appears that this suggestion has not been taken into account in the revised version.
Author Response
The authors have satisfactorily addressed all my previous concerns regarding the Discussion section. However, I still believe that including a summarizing figure would help highlight the key results and mechanistic insights. It appears that this suggestion has not been taken into account in the revised version.
We thank the reviewer for this helpful reminder. A final summarising figure has now been included to highlight the key findings and proposed mechanistic interpretation (Figure 8). Please, see page 24.
Reviewer 4 Report
Comments and Suggestions for Authors
The revised version is acceptable
Author Response
The revised version is acceptable
Thank you to the reviewer for their relevant review of the manuscript.
Round 3
Reviewer 1 Report
Comments and Suggestions for Authors
The revised version satisfactorily addresses all major concerns raised in the previous review. The authors provided clear methodological clarifications, added appropriate controls, corrected terminology, and reanalyzed data using mixed models with FDR correction. The Discussion now reflects a balanced interpretation of the findings, acknowledging methodological limitations and avoiding causal overstatements. The manuscript is substantially improved in clarity, rigor, and consistency.